# Train with Perturbation, Infer after Merging:
# A Two-Stage Framework for Continual Learning

**Haomiao Qiu**[1,2]**, Miao Zhang**[1]\***, Ziyue Qiao**[2]\***,  Liqiang Nie**[1]

[1] Harbin Institute of Technology (Shenzhen)
[2] Great Bay University
`24B951058@stu.hit.edu.cn, zhangmiao@hit.edu.cn, zyqiao@gbu.edu.cn,`
`nieliqiang@gmail.com`

## Abstract

Continual Learning (CL) aims to enable models to continuously acquire new knowledge from a sequence of tasks with avoiding the forgetting of learned information. However, existing CL methods only rely on the parameters of the most recent task for inference, which makes them susceptible to catastrophic forgetting. Inspired by the recent success of model merging techniques, we propose **Perturb-and-Merge (P&M)**, a novel continual learning framework that integrates model merging into the CL paradigm to mitigate forgetting. Specifically, after training on each task, P&M constructs a new model by forming a convex combination of the previous model and the newly trained task-specific model. Through theoretical analysis, We minimize the total loss increase across all tasks and derive a closed-form solution for the merging coefficient under mild assumptions. To further improve the performance of the merged model, we observe that the degradation introduced during merging can be alleviated by a regularization term composed of the task vector and the Hessian matrix of the loss function. Interestingly, we show that this term can be efficiently approximated using second-order symmetric finite differences, and a stochastic perturbation strategy along the task vector direction is accordingly devised which incurs no additional forward or backward passes while providing an effective approximation of the regularization term. Finally, we combine P&M with LoRA, a parameter-efficient fine-tuning method, to reduce memory overhead. Our proposed approach achieves state-of-the-art performance on several continual learning benchmark datasets. The code is available at `https://github.com/qhmiao/P-M-for-Continual-Learning`.

## 1 Introduction

Continual Learning (CL) aims to enable models to continuously acquire new knowledge from a sequence of tasks while retaining previously learned information [1]. In many real-world applications, data arrives in a streaming fashion, and due to constraints such as privacy, storage, or computational resources, models are typically unable to retain or revisit earlier task data. This setting poses a fundamental challenge: how can a model continually adapt to new tasks while maintaining good performance on previous ones? Although considerable progress has been made through methods such as parameter isolation [2, 3, 4], regularization [5, 6, 7, 8], and experience replay [9, 10, 11, 12], catastrophic forgetting remains a core issue in CL.

In parallel, the rise of large-scale pretrained models has spurred interest in model merging as a simple yet effective strategy for post-training enhancement. Prior studies have shown that independently

---

*Co-corresponding Authors.

39th Conference on Neural Information Processing Systems (NeurIPS 2025).

trained models, initialized from the same pretrained weights and optimized on different tasks, can be effectively merged—through parameter interpolation or more structured methods—into a single unified model that performs well across tasks [13, 14, 15, 16, 17, 18, 19, 20]. Model merging offers a practical mechanism for consolidating task-specific models without requiring shared training data.

Although CL and model merging differ significantly in their procedures, they share a common goal: to learn a single model that performs well across multiple tasks. The key distinction lies in the assumptions and timing of parameter updates: CL follows a sequential paradigm, where only current-task data is accessible and previous task parameters are reused for initialization; whereas model merging assumes that tasks are trained independently from the same pretrained model, without any data sharing. Both paradigms operate under task-isolated settings but differ in how and when task integration occurs.

A major advantage of CL is that all tasks are trained along a shared optimization trajectory, which increases the likelihood that the resulting parameters reside near a joint optimum [21, 22]. In contrast, model merging offers a stable post-training integration mechanism, particularly attractive in scenarios where previous data cannot be revisited. Notably, most CL methods use the model parameters obtained after task $t$ to perform inference on all tasks 1 to $t$, even though these parameters are mainly optimized for the current task [23, 21, 24].

Motivated by this observation, we propose a novel method, **Perturb-and-Merge (P&M)**, which unifies the training dynamics of CL with the inference principle of model merging.

**Infer after Merging.** Specifically, for task $t$, we initialize training with the inference parameters $\hat{\theta}_{t-1}$ from previous tasks, and after training, obtain the task-specific optimum $\theta_t^*$. We then merge the two using the convex combination: $\hat{\theta}_t = (1 - \alpha)\hat{\theta}_{t-1} + \alpha\theta_t^*$. This process can be viewed as scaling the task vector $\Delta\theta_t^* = \theta_t^* - \hat{\theta}_{t-1}$ to reduce interference with previously learned knowledge. To determine the optimal merging coefficient $\alpha$, we analyze its influence on the total loss across all tasks and derive a closed-form solution that minimizes overall performance degradation. This solution depends on the Hessian matrix at the optimum of each task, which we approximate using the empirical Fisher information matrix.

**Train with Perturbation.** Furthermore, because model merging can introduce parameter conflicts—causing the merged model to underperform compared to task-specific models—we find that this degradation can be mitigated by introducing a regularization loss term composed of the task vector and the Hessian matrix during training. However, computing the Hessian matrix on a per-batch basis is computationally expensive. Interestingly, we show that this regularization term can be efficiently approximated using second-order symmetric finite differences. More importantly, injecting task-vector-aligned parameter perturbations during training provides a stochastic approximation of the regularizer, requiring no additional forward or backward passes. Our experiments demonstrate that such perturbations can effectively reduce parameter conflicts during model merging and enhance the performance of the merged model.

Finally, we combine P&M with LoRA (Low-Rank Adaptation) [25], a parameter-efficient fine-tuning strategy, to reduce memory overhead for model storage. Our overall approach achieves state-of-the-art performance on multiple CL benchmark datasets.

## 2 Related Work

### 2.1 Continual Learning

CL aims to enable models to learn new tasks sequentially without forgetting previously acquired knowledge. Regularization-based methods preserve important parameters by applying constraints to prevent forgetting [5, 6, 7, 8], while memory-based approaches use external buffers to store historical data for rehearsal or sampling [9, 10, 11, 12]. Architecture-based methods dynamically expand model capacity to accommodate new tasks [2, 3, 4]. Another direction focuses on constraining gradient directions to reduce task interference. Examples include Orthogonal Weight Modulation (OWM)[26], Orthogonal Gradient Descent (OGD)[27], and Gradient Projection Memory (GPM) [28], which project gradients into task-specific subspaces to retain prior knowledge.

With the rise of large-scale pre-trained models [29, 30, 31], continual fine-tuning has become increasingly popular. However, full fine-tuning is computationally expensive, and more parameter-

efficient tuning strategies—such as prompt-based learning [32, 33, 34], LoRA, and modular tuning methods [35]—have been proposed to improve the scalability and practicality of CL. In this work, we integrate our method with LoRA to reduce both training and storage costs.

## 2.2 Model Merging

Model merging has gained traction in both multi-task and CL scenarios. Generally, it can be categorized into two directions: one line of work merges models fine-tuned on the same task to improve generalization [36, 37, 38, 39]; the other merges models trained on different tasks to form a single, unified model that can handle all constituent tasks. Model merging aims to integrate knowledge from different tasks without retraining from scratch [40, 41, 42]. Simple strategies use fixed merging weights based on the number of tasks [43], while more advanced methods such as IMM [44] and CoMA [22] empirically tune coefficients to improve performance.

Some approaches [45, 39] go further by assigning parameter-wise merging coefficients to reflect the heterogeneous impact of different weights, but such methods often require extensive hyperparameter tuning. In contrast, our method provides a closed-form, theoretically optimal merging strategy without the need for manual tuning.

# 3 Method

## 3.1 Infer after Merging

In continual learning, there are $T$ tasks $\mathcal{T}_1, ..., \mathcal{T}_T$ that each task includes data: $\mathcal{D}_t = \{(\mathbf{x}_i^t, y_i^t)\}_{i=1}^{n_t}$, where $\mathbf{x}_i^t \in \mathbb{R}^d$ is the input, $y_i^t \in \mathbb{R}$ is the label, and $f_{\theta_0}(\cdot)$ is a pretrain model with parameters $\theta_0$.

When receiving the $t$-th task, the goal of CL is to achieve an overall optimal performance across all $T$ tasks. Since we currently only have access to data from task $t$, we can only update $\theta_t$ by training on $\mathcal{D}_t$. Given an input $\mathbf{x}$ with ground-truth label $y$ from $\mathcal{D}_t$ and a model prediction $\hat{\mathbf{p}} = \mathrm{softmax}(f_\theta(\mathbf{x})) \in \mathbb{R}^C$, the cross-entropy loss is defined as:

$$\mathcal{L}^{\mathrm{ce}}(\theta) = -\sum_{c=1}^{C} \mathbb{I}(y = c) \log \hat{p}_c = -\log \hat{p}_y, \tag{1}$$

where $\hat{p}_y$ is the predicted probability for the correct class $y$ and $\mathbb{I}(\cdot)$ is the indicator function. At this point, we have the model $\hat{\theta}_{t-1}$ used for inference on tasks 1 to $t-1$, which we take as the starting point for training. After training with Eq. 1, we obtain the optimal parameters $\theta_t^*$ on the dataset $\mathcal{D}_t$:

$$\theta_t^* = \hat{\theta}_{t-1} + \Delta\theta_t^* = \hat{\theta}_{t-1} + \underset{\Delta\theta_t}{\mathrm{argmin}}\, \mathcal{L}_t(\hat{\theta}_{t-1} + \Delta\theta_t). \tag{2}$$

Traditional CL methods directly use $\theta_t^*$ as the inference parameter for the $t$-th task, i.e., $\hat{\theta}_t = \theta_t^*$. However, since $\theta_t^{l*}$ is trained on $\mathcal{D}_t$, it may not fully guarantee the performance on $\mathcal{D}_1, ..., \mathcal{D}_{t-1}$.

A natural idea is to combine models trained on different tasks. Specifically, $\hat{\theta}_{t-1}$, which be used as the inference parameter for these tasks, performs well on tasks 1 to $t - 1$, while $\theta_t^*$, which is optimized for task $t$, serves as the inference parameter for task $t$. Therefore, we can merge the two models. Here we use a simple convex combination:

$$\hat{\theta}_t = (1 - \alpha_t)\hat{\theta}_{t-1} + \alpha_t \theta_t^*, \tag{3}$$

where $0 \le \alpha_t \le 1$ is the merging coefficient used to balance the weights of the two parameters, and $\hat{\theta}_t$ is used for inference on tasks 1 to $t$. Since $\theta_t^*$ is trained based on $\hat{\theta}_{t-1}$, i.e., $\theta_t^* = \hat{\theta}_{t-1} + \Delta\theta_t^*$, then the Eq. 3 can be rewritten as:

$$\hat{\theta}_t = \hat{\theta}_{t-1} + \alpha_t \Delta\theta_t^*, \tag{4}$$

which means that taking the weighted average of $\hat{\theta}_{t-1}$ and $\theta_t^*$ is equivalent to scaling $\Delta\theta_t^*$. This scaling strategy means that it will not harm the parameters $\hat{\theta}_{t-1}$ for tasks 1 to $t - 1$, but only by adjusting the task vector for task $t$ to reduce forgetting of old tasks. In our experiments 4.3, we found that this scaling has a slight impact on the knowledge of the new task.

Table 1: Training and Inference Strategies

| Task $t$ | Training: $\theta_t^* = argmin_{\theta_t} \mathcal{L}_t(\theta_t)$ | Inference: $\hat{\theta}_t$ |
|---|---|---|
| Continual Learning | $\theta_t^* = \theta_{t-1} + \Delta\theta_t^*$ | $\hat{\theta}_t = \theta_t^*$ |
| Model Merging | $\theta_t^* = \theta_0 + \Delta\theta_t^*$ | $\hat{\theta}_t = \sum_{i=1}^t \alpha_i \theta_i^*$ |
| Ours (P&M) | $\theta_t^* = \theta_{t-1} + \Delta\theta_t^*$ | $\hat{\theta}_t = \hat{\theta}_{t-1} + \alpha_t \Delta\theta_t^*$ |

**Infer after Merging** unifies the benefits of model merging and continual learning, as demonstrated in Tab. 1. In contrast to conventional CL, it mitigates interference with previously acquired knowledge while encouraging more compact task-specific optima through continued adaptation—ultimately promoting better generalization [21, 46]. Unlike momentum updates or exponential moving averages (EMA) [47] [48] that perform step-wise smoothing, our method merges the full task vector after completing each task. Instead of relying on historical gradients or parameter trajectories, it directly combines the current task's optimal solution with the previous model, better preserving task semantics and overall structure.

## 3.2 A Closed-form Solution for Optimal Merge Coefficient

Next, we aim to obtain an optimal $\alpha_t$. We are concerned with the performance degradation of the merged model $\hat{\theta}_t$ compared to each task-specific optimal model $\theta_i^*$, where $i \leq t$. For task $i$, we define the performance drop $\delta_i$ as:

$$\delta_i = \mathcal{L}_i(\hat{\theta}_t) - \mathcal{L}_i(\theta_i^*). \tag{5}$$

$\delta_i$ evaluates the impact of the merged model $\hat{\theta}_t$ on the loss of task $i$ compared to its optimal parameter $\theta_i^*$. We expand it into Taylor series, then we have

$$\delta_i = \nabla\mathcal{L}_i(\theta_i^*)(\hat{\theta}_t - \theta_i^*) + \frac{1}{2}(\hat{\theta}_t - \theta_i^*)^\top \mathbf{H}_i(\theta_i^*)(\hat{\theta}_t - \theta_i^*) + O(\|\hat{\theta}_t - \theta_i^*\|^3), \tag{6}$$

where $\mathbf{H}_i(\theta_i^*)$ represents the corresponding Hessian matrix. We can consider that for task $i$, $\theta_t^*$ is the optimal parameter for task $i$, so $\nabla\mathcal{L}_i(\theta_i^*)$ tends to 0. And We can assume that $\|\hat{\theta}_t - \theta_i^*\|^3$ tends to 0, so we only need to consider the second-order term $\frac{1}{2}(\hat{\theta}_t - \theta_i^*)^\top \mathbf{H}_i(\theta_i^*)(\hat{\theta}_t - \theta_i^*)$. Then we consider the change in the loss for all tasks, i.e.,

$$\alpha_t^* = \underset{\alpha_t}{\operatorname{argmin}} \sum_{i=1}^t \delta_i \approx \underset{\alpha_t}{\operatorname{argmin}} \sum_{i=1}^t \frac{1}{2}(\hat{\theta}_t - \theta_i^*)^\top \mathbf{H}(\theta_i^*)(\hat{\theta}_t - \theta_i^*), \tag{7}$$

which has the closed-form solution:

$$\alpha_t^* = -\frac{\sum_{i=1}^t \left(\hat{\theta}_{t-1} - \theta_i^*\right)^\top \mathbf{H}_i(\theta_i^*)\Delta\theta_t^*}{\sum_{i=1}^t \Delta\theta_t^* \mathbf{H}_i(\theta_i^*)\Delta\theta_t^*}, \tag{8}$$

Please refer to the Appendix A.1 for the detailed derivation. To avoid the prohibitive cost of computing the full Hessian, we approximate it with the diagonal of the empirical Fisher information matrix [49, 50]. Given a model with parameters $\theta$, the empirical Fisher matrix is defined as:

$$\mathbf{F}(\theta) = \mathbb{E}_{(\mathbf{x},y)\sim\mathcal{D}} \left[\nabla_\theta \log p_\theta(y \mid \mathbf{x})\nabla_\theta \log p_\theta(y \mid \mathbf{x})^\top\right]. \tag{9}$$

In practice, we approximate $\mathbf{F}(\theta)$ using the average over the whole training dataset:

$$\hat{\mathbf{F}}(\theta) = \frac{1}{N} \sum_{i=1}^N \nabla_\theta \log p_\theta(y_i \mid \mathbf{x}_i)\nabla_\theta \log p_\theta(y_i \mid \mathbf{x}_i)^\top, \tag{10}$$

and use only the diagonal entries of $\hat{\mathbf{F}}(\theta)$. After completing the training of task $t$, we first compute the diagonal empirical Fisher matrix $\hat{\mathbf{F}}_t(\theta_t^*) \approx -\mathbf{H}_t(\theta_t^*)$, then obtain $\alpha_t^*$ using Eq. 8, and finally compute the merged model $\hat{\theta}_t = \hat{\theta}_{t-1} + \alpha_t^*\Delta\theta_t^*$ as the inference parameters for tasks 1 to $t$. Note that for each previous task $i < t$, the corresponding Fisher matrix $\hat{\mathbf{F}}_i(\theta_i^*)$ is computed and stored at the time of training task $i$.

### 3.3 Train with Perturbation

Further, we can also reduce $\sum_{i=1}^{t} \delta_i(\alpha_t^*)$ to make the merged model optimal on all $t$ tasks by optimizing $\theta_t^*$. Next, we abbreviate $\mathbf{H}_i(\theta_i^*)$ as $\mathbf{H}_i$, then we have

$$\sum_{i=1}^{t} \delta_i(\alpha_t^*) = \frac{1}{2} \sum_{i=1}^{t} \left( \hat{\theta}_{t-1} - \theta_i^* \right)^{\top} \mathbf{H}_i \left( \hat{\theta}_{t-1} - \theta_i^* \right) - \frac{\left( \sum_{i=1}^{t} \left( \hat{\theta}_{t-1} - \theta_i^* \right)^{\top} \mathbf{H}_i \Delta\theta_t^* \right)^2}{2 \sum_{i=1}^{t} \Delta\theta_t^{*\top} \mathbf{H}_i \Delta\theta_t^*} \quad (11)$$

$$\leq \frac{1}{2} \sum_{i=1}^{t} \left( \hat{\theta}_{t-1} - \theta_i^* \right)^{\top} \mathbf{H}_i \left( \hat{\theta}_{t-1} - \theta_i^* \right) \quad (12)$$

For the details, please refer to the Appendix A.2. To enhance the performance of the merged model, we aim to minimize the upper bound of $\sum_{i=1}^{t} \delta_i(\alpha_t^*)$. Among all the terms in this expression, only the last term $\delta_t(\alpha_t^*)$ involving $\theta_t^*$—specifically, $\left( \hat{\theta}_{t-1} - \theta_t^* \right)^{\top} \mathbf{H}_t \left( \hat{\theta}_{t-1} - \theta_t^* \right) = \Delta\theta_t^{*\top} \mathbf{H}_t \Delta\theta_t^*$—depends on the current task $t$ and can be optimized during training.

During training, this term can be added as a regularizer to reduce $\delta_t(\alpha_t^*)$, but it requires real-time Hessian computation. While the Fisher Information Matrix is commonly used as a surrogate, it only approximates the Hessian near the optimum under certain conditions (e.g., the number of data samples $N \longrightarrow \infty$). Since training parameters are far from optimal and the Fisher can only be estimated from a single batch during training, it fails to capture the true curvature and is thus unreliable in this setting. Fortunately, we can approximate the quadratic form using symmetric finite differences as follows:

$$\Delta\theta_t^{\top} \mathbf{H}_t \Delta\theta_t \approx \frac{1}{\epsilon^2} \left( \mathcal{L}_t^{\text{ce}}(\theta_t + \epsilon\Delta\theta_t) + \mathcal{L}_t^{\text{ce}}(\theta_t - \epsilon\Delta\theta_t) - 2\mathcal{L}_t^{\text{ce}}(\theta_t) \right), \quad (13)$$

where $\epsilon \leq 1$ is a small constant, $\Delta\theta_t$ denotes the task vector during training and $\theta_t = \hat{\theta}_{t-1} + \Delta\theta_t$. This approximation is derived via a second-order Taylor expansion of the loss around $\theta$. We incorporate the right-hand side of Eq. 13 as a regularization term during training, and define the total training loss as:

$$\mathcal{L}_t(\theta_t) = \mathcal{L}_t^{\text{ce}}(\theta_t) + \lambda \mathcal{L}_t^{\text{reg}}(\theta_t)$$
$$= \left( 1 - \frac{2\lambda}{\epsilon^2} \right) \mathcal{L}_t^{\text{ce}}(\theta_t) + \frac{\lambda}{\epsilon^2} \left( \mathcal{L}_t^{\text{ce}}(\theta_t + \epsilon\Delta\theta_t) + \mathcal{L}_t^{\text{ce}}(\theta_t - \epsilon\Delta\theta_t) \right), \quad (14)$$

where $\lambda$ control the strength of the regularization term. Eq. 14 requires two additional forward passes of the cross-entropy loss with parameter perturbations along the task vector direction in each training step. While this avoids computing the full Hessian, it results in a threefold increase in memory and computation cost. To improve efficiency, we propose a stochastic approximation.

We observe that Eq. 14 is effectively equivalent to applying a random perturbation to the model parameters during training, where the perturbation direction aligns with the task vector of task $t$. Specifically, the loss is evaluated under one of three perturbations: $\epsilon\Delta\theta_t$, $-\epsilon\Delta\theta_t$, or $0$. Specifically, at each training step, we sample one of the three versions of the loss:

$$\tilde{\mathcal{L}}_t(\theta_t) = \begin{cases} \mathcal{L}_t^{\text{ce}}(\theta_t) & \text{with probability } p_0, \\ \mathcal{L}_t^{\text{ce}}(\theta_t + \epsilon\Delta\theta_t^*) & \text{with probability } p_+, \\ \mathcal{L}_t^{\text{ce}}(\theta_t - \epsilon\Delta\theta_t^*) & \text{with probability } p_-. \end{cases} \quad (15)$$

We define the sampling probabilities for the three perturbation terms as follows: the original point is sampled with probability $p_0 = 1 - \frac{2\lambda}{\epsilon^2}$, and the two perturbed points are each sampled with probability $p_+ = p_- = \frac{\lambda}{\epsilon^2}$. In the experiments, we treat $p_0$ as a hyperparameter, and set $p_+ = p_- = \frac{1}{2}(1 - p_0)$. These probabilities are designed to ensure that the expected loss remains consistent with the original definition, resulting in an unbiased estimate of the total loss:

$$\mathbb{E}[\tilde{\mathcal{L}}_t(\theta)] = p_0 \mathcal{L}_t^{\text{ce}}(\theta) + p_+ \mathcal{L}_t^{\text{ce}}(\theta + \epsilon\Delta\theta_t^*) + p_- \mathcal{L}_t^{\text{ce}}(\theta - \epsilon\Delta\theta_t^*) = \mathcal{L}_t(\theta). \quad (16)$$

This sampling strategy reduces the forward cost per batch from 3× to 1×, without introducing bias into the gradient estimation. While variance may increase slightly, this technique enables us to scale P&M to large models and datasets efficiently.

**Algorithm 1** LoRA-P&M

---

1: **Input:** Datasets $\mathcal{D}_t = \{(\mathbf{x}_i^t, y_i^t)\}_{i=1}^{n_t}$, for $T$ tasks $\mathcal{T}_1, ..., \mathcal{T}_T$, a pre-trained model $f_\theta(\cdot)$ with parameters $\theta_0$, $\hat{\theta}_0 = \theta_0$, hyper-parameters $\epsilon$, $p_0$ and $p_+ = p_- = \frac{1}{2}(1 - p_0)$.

2: **Output:** $\hat{\theta}_T$ for all tasks.

3: **for** $t = 1$ to $T$ **do**

4:     Fix $\hat{\theta}_{t-1}$ and initialize a new task-specific LoRA set for task t: $\text{LoRA}_t$

5:     **repeat**

6:         **for** each example $(\mathbf{x}_i^t, y_i^t) \in \mathcal{D}_t$ **do**

7:             ▷ Train with Perturbation:

8:             Sample perturbation $\tilde{\epsilon} \in \{-\epsilon, 0, +\epsilon\}$ according to probabilities $\{p^-, p_0, p^+\}$

9:             Compute loss $\mathcal{L}_t^{\text{ce}}(\hat{\theta}_{t-1} + (1 + \tilde{\epsilon})\text{LoRA}_t; \mathbf{x}_i^t, y_i^t)$

10:             Update $\text{LoRA}_t$ via AdamW

11:         **end for**

12:     **until** convergence

13:     ▷ Infer after Merging:

14:     $\theta_t^* = \hat{\theta}_{t-1} + \text{LoRA}_t$

15:     Estimate the Fisher Information Matrix $\hat{\mathbf{F}}_t(\theta_t^*)$ on task $\mathcal{T}_t$ using Eq. 10 and Alg. 2

16:     $\hat{\theta}_t = \hat{\theta}_{t-1} + \alpha_t^*\text{LoRA}_t$ where $\alpha_t^*$ is from Eq. 8

17: **end for**

---

### 3.4 LoRA-P&M

Note that computing Eq. 8 requires storing the optimal parameters $\theta_i^*$ of all previous tasks. As the number of tasks grows, this results in increased memory demands. To address this issue, we integrate LoRA [25], a low-rank parameter-efficient fine-tuning method, to reduce storage overhead.

For task $t$, the linear layer becomes $\mathbf{W}_t = \mathbf{W}_{t-1} + \mathbf{A}_t\mathbf{B}_t$, where only $\mathbf{A}_t$ and $\mathbf{B}_t$ are updated. LoRA reduces trainable parameters by decomposing the weight update into low-rank matrices: $\Delta\mathbf{W} = \mathbf{AB}$. In CL, new LoRA module will be added for new task $t$, while other modules are kept fixed and participate jointly in the forward pass [23, 51].

Let $\mathcal{P}_{\text{LoRA}} = \{(\mathbf{A}^{(p)}, \mathbf{B}^{(p)}) \mid p \in \mathcal{I}\}$ denote the set of LoRA modules inserted at various locations $p$ in the network, where $\mathcal{I}$ indexes the parameter subsets affected. The overall model parameters $\theta_t$ can then be written as:

$$\theta_t = \theta_{t-1} + \bigoplus_{p \in \mathcal{P}} \left(A_t^{(p)} B_t^{(p)}\right), \tag{17}$$

$\oplus$ denotes location-wise module insertion, not element-wise addition.

In our experiments, we apply LoRA modules to the key and value projections, so that only the Fisher Information Matrices of these parameters need to be stored. The experiments comparing the memory and time overhead of our method with LoRA are provided in the Appendix B.3. The overall procedure of the algorithm is presented in Algorithm 1.

## 4 Experiments

In this section, we first present the experimental setups, and then compare P&M with state-of-the-art CL methods and model merging methods across multiple benchmarks.

### 4.1 Experimental Setups

**Evaluation Benchmarks and Metrics.** Following the evaluation protocols in [35, 23], we assess LoRA-P&M on five standard CL benchmarks: ImageNet-R [52], ImageNet-A [53], DomainNet [54], CIFAR100 [55], and CUB200 [56]. As in prior work [23, 21], we split ImageNet-R into 5, 10, and

Table 2: Performance comparison with CL methods on ImageNet-R across different task lengths.

| Method | ImageNet-R 5 tasks | | ImageNet-R 10 tasks | | ImageNet-R 20 tasks | |
|---|---|---|---|---|---|---|
| | Acc ↑ | AAA ↑ | Acc ↑ | AAA ↑ | Acc ↑ | AAA ↑ |
| Full Fine-Tuning | 64.92± 0.87 | 75.57± 0.50 | 60.57± 1.06 | 72.31± 1.09 | 49.95± 1.31 | 65.32± 0.84 |
| L2P [33] | 73.04± 0.71 | 76.94± 0.41 | 71.26± 0.44 | 76.13± 0.46 | 68.97± 0.51 | 74.16± 0.32 |
| DualPrompt [58] | 69.99± 0.57 | 72.24± 0.41 | 68.22± 0.20 | 73.81± 0.39 | 65.23± 0.45 | 71.30± 0.16 |
| CODA-Prompt [32] | 76.63± 0.27 | 80.30± 0.28 | 74.05± 0.41 | 78.14± 0.39 | 69.38± 0.33 | 73.95± 0.63 |
| HiDe-Prompt [62] | 74.77± 0.25 | 78.15± 0.24 | 74.65± 0.14 | 78.46± 0.18 | 73.59± 0.19 | 77.93± 0.19 |
| InfLoRA [23] | 76.95± 0.23 | 81.81± 0.14 | 74.75± 0.64 | 80.67± 0.55 | 69.89± 0.56 | 76.68± 0.57 |
| SD-LoRA [21] | 79.15± 0.20 | 83.01± 0.42 | 77.34± 0.35 | 82.04± 0.24 | 75.26± 0.37 | 80.22± 0.72 |
| LoRA | 71.22± 1.47 | 78.15± 1.08 | 65.72± 0.75 | 76.14± 0.96 | 56.35± 0.80 | 71.08± 1.04 |
| **LoRA-P&M** | **81.47 ± 0.56** | **85.96 ± 0.52** | **79.95 ± 0.18** | **85.29 ± 0.93** | **76.37 ± 0.09** | **82.77 ± 0.71** |

Table 3: Performance comparison with model merging methods on 5 datasets.

| Method | INR-10 | INR-20 | INA-10 | DN*-5 | C100-10 | CUB-10 |
|---|---|---|---|---|---|---|
| | Acc ↑ | Acc ↑ | Acc ↑ | Acc ↑ | Acc ↑ | Acc ↑ |
| LoRA | 65.72 | 56.35 | 44.41 | 71.81 | 72.58 | 64.82 |
| w/ Model Averaging | 76.90 | 74.64 | 54.54 | 81.84 | 87.52 | 74.87 |
| w/ DARE [39] | 75.09 | 66.03 | 55.87 | 80.58 | 87.28 | 76.57 |
| w/ CoMA [63] | 79.34 | 75.60 | 53.24 | 83.98 | 86.95 | 74.65 |
| w/ CoFIMA [63] | 79.06 | 75.09 | 54.09 | 83.85 | 86.58 | 74.43 |
| w/ **P&M** | **79.95** | **76.37** | **56.57** | **84.71** | **88.45** | **78.29** |

20 tasks; ImageNet-A into 10 tasks; DomainNet into 5 tasks; and both CIFAR100 and CUB200 into 10 tasks. Specifically, *DomainNet* refers to the full version with all 345 classes, while *DN\** denotes a variant where we select the top 200 most populous classes, following [57, 54].

Following [21], we report two widely used CL metrics: average accuracy (Acc) and average anytime accuracy (AAA). Acc computes the mean accuracy over all $N$ tasks after all tasks is completed. AAA further captures learning dynamics by averaging accuracy on all seen tasks after training on each new task.

**Competing Methods and Implementation Details.** We compare LoRA-P&M against state-of-the-art ViT-based CL methods, including L2P [33], DualPrompt [58], CODA-Prompt [32], HiDe-Prompt [59], InfLoRA [23] and SD-LoRA [21], while full fine-tuning as a form of performance lower bound. We also compare against model merging methods, including model averaging, DARE [39], CoMA [22] and CoFIMA [22]. Following prior work [35], we employ ViT-B/16 [30], pre-trained on ImageNet-21K and fine-tuned on ImageNet-1K as the foundation model for classification. Following [23], we insert LoRA (rank=10) modules into the key and value projections in multi-head attention. We set $\epsilon = 0.5$ in Eq. 15, with uniform sampling $p_0 = p_+ = p_- = \frac{1}{3}$. Our method is optimized using AdamW [60] with an initial learning rate of $1e-3$ for LoRA and $1e-2$ for the classification head following [61]. We use a batch size of 256 across all datasets. Each task is trained for 10 epochs, except for DomainNet, which is trained for 5 epochs. We report the mean and standard deviation over three runs to reflect the stability of the results. All results are obtained by running on a single NVIDIA L40s GPU.

## 4.2 Main Results

**Comparison with State-of-the-Art CL Methods.** As shown in Tab. 2, **LoRA-P&M** consistently outperforms both the original LoRA baseline and recent CL methods, including SD-LoRA [21], across all evaluated settings. On ImageNet-R with 5, 10, and 20 tasks, our method yields significant gains over LoRA (e.g., +14.23% in the 10-task setting) and surpasses SD-LoRA by up to +2.61%. For additional results on DomainNet, ImageNet-A, CIFAR100, and CUB200, please refer to Appendix B.

**Comparison with Model Merging Methods.** Tab. 3 compares LoRA-P&M with LoRA and several representative model merging methods across six benchmarks. P&M delivers substantial improvements over LoRA—up to +13.6% on INR-20 and +11.1% on C100-10—highlighting the benefits of post-training merging in preserving task knowledge. While methods such as DARE and CoFIMA offer moderate improvements over LoRA, they consistently underperform compared to P&M. Notably, P&M exceeds CoFIMA by +3.86% on CUB-10 and +2.48% on INA-10.

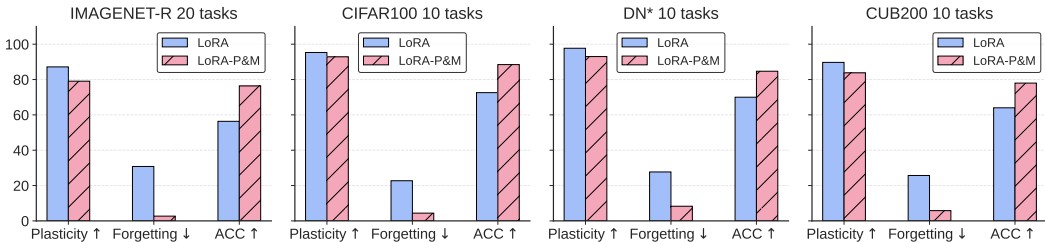

Figure 1: **P&M reduces forgetting with minimal impact on plasticity.** Comparison of LoRA and LoRA-P&M across four benchmarks. P&M achieves similar plasticity (current task performance) while significantly mitigating forgetting (average performance drop on previous tasks), resulting in higher overall ACC.

Table 4: Performance comparison on ImageNet-A and DomainNet.

| Method | ImageNet-A 10 tasks | | DomainNet 5 tasks | |
| --- | --- | --- | --- | --- |
| | Acc ↑ | AAA ↑ | Acc ↑ | AAA ↑ |
| Full Fine-Tuning | $16.31 \pm 7.89$ | $30.04 \pm 13.18$ | $51.46 \pm 0.47$ | $67.08 \pm 1.13$ |
| L2P [33] | $42.94 \pm 1.27$ | $51.40 \pm 1.95$ | $70.26 \pm 0.25$ | $75.83 \pm 0.98$ |
| DualPrompt [58] | $45.49 \pm 0.96$ | $54.68 \pm 1.24$ | $68.26 \pm 0.90$ | $73.84 \pm 0.45$ |
| CODA-Prompt [32] | $45.36 \pm 0.78$ | $57.03 \pm 0.94$ | $70.58 \pm 0.53$ | $76.68 \pm 0.44$ |
| HiDe-Prompt [59] | $42.70 \pm 0.60$ | $56.32 \pm 0.40$ | $72.20 \pm 0.08$ | $77.01 \pm 0.04$ |
| InfLoRA [23] | $49.20 \pm 1.12$ | $60.92 \pm 0.61$ | $71.59 \pm 0.23$ | $78.29 \pm 0.50$ |
| SDLoRA [21] | $55.96 \pm 0.73$ | $64.95 \pm 1.63$ | $72.82 \pm 0.37$ | $78.89 \pm 0.50$ |
| SD-LoRA-RR [21] | $55.59 \pm 1.08$ | $64.59 \pm 1.91$ | $72.58 \pm 0.40$ | $78.79 \pm 0.78$ |
| SD-LoRA-KD [21] | $54.24 \pm 1.12$ | $63.89 \pm 0.58$ | $72.15 \pm 0.50$ | $78.44 \pm 0.66$ |
| LoRA | $44.41 \pm 0.57$ | $56.76 \pm 3.54$ | $64.82 \pm 2.30$ | $75.28 \pm 1.56$ |
| LoRA-P&M | $\mathbf{56.57 \pm 0.78}$ | $\mathbf{65.35 \pm 1.81}$ | $\mathbf{76.91 \pm 0.53}$ | $\mathbf{85.27 \pm 0.07}$ |

**More results on DomainNet, ImageNet-A, CIFAR100, and CUB200.** As in shown in Tab. 4 and 5 Across all four benchmarks, LoRA-P&M consistently outperforms both standard LoRA and the recent SD-LoRA. On challenging datasets such as ImageNet-A and DomainNet, it achieves the highest accuracy and AAA, surpassing SD-LoRA by up to +0.61% and +6.38%, respectively. Notably, on CIFAR100 and CUB200, LoRA-P&M matches or exceeds SD-LoRA's accuracy while maintaining lower variance. These results highlight the robustness and generalization advantage of post-training task vector merging over both naive LoRA and structured dynamic variants like SD-LoRA.

**Ablation on the Merging Coefficient Strategy.** A potential concern is that using a uniform merging coefficient $\alpha$ across all parameters might be overly restrictive. To evaluate this, we compared our global-$\alpha$ strategy with a more fine-grained variant that learns separate $\alpha$ values for each LoRA module. The results on ImageNet-R are summarized in Table 6. The performance differences are negligible. We further observed that the learned per-module $\alpha$ values were highly similar to each other, suggesting that a single global coefficient is sufficient to capture the merging dynamics. Therefore, we adopt the global-$\alpha$ strategy for its simplicity, efficiency, and theoretical tractability. This finding aligns with prior research [13], where a unified coefficient was shown to preserve the structural integrity of the task vector $\Delta\theta_t$, which represents a coherent direction in parameter space. In contrast, a parameter-wise $\alpha$ may introduce inconsistent scaling and distort the internal correlations of $\Delta\theta_t^*$, thereby complicating the derivation of closed-form solutions.

## 4.3 Analysis

This subsection investigates how **P&M** enhances CL performance.

**Observation ①: Scaling task vectors reduces forgetting with minimal impact on plasticity.** We define plasticity as the average accuracy on the new task and forgetting as the average accuracy drop on previously learned tasks. As shown in Fig. 1, scaling the task vector during inference (**P&M**)

Table 5: Performance comparison on CIFAR100 and CUB200.

| Method | CIFAR100 10 tasks | | CUB200 10 tasks | |
|---|---|---|---|---|
| | Acc ↑ | AAA ↑ | Acc ↑ | AAA ↑ |
| Full Fine-Tuning | $69.49 \pm 0.50$ | $80.35 \pm 0.87$ | $51.43 \pm 1.41$ | $69.74 \pm 0.93$ |
| L2P [33] | $83.18 \pm 1.20$ | $87.69 \pm 1.05$ | $65.18 \pm 2.49$ | $76.12 \pm 1.27$ |
| DualPrompt [58] | $81.48 \pm 0.86$ | $86.41 \pm 0.66$ | $68.00 \pm 1.06$ | $79.40 \pm 0.88$ |
| CODA-Prompt [32] | $86.31 \pm 0.12$ | $90.67 \pm 0.22$ | $71.92 \pm 0.33$ | $78.76 \pm 0.65$ |
| InfLoRA [23] | $86.75 \pm 0.35$ | $91.72 \pm 0.15$ | $70.82 \pm 0.23$ | $81.39 \pm 0.14$ |
| SD-LoRA [21] | $88.01 \pm 0.31$ | $92.54 \pm 0.18$ | $77.48 \pm 0.20$ | $\mathbf{85.59 \pm 0.44}$ |
| LoRA | $72.58 \pm 1.57$ | $80.64 \pm 2.31$ | $64.82 \pm 2.30$ | $75.28 \pm 1.56$ |
| LoRA-P&M | $\mathbf{88.45 \pm 0.35}$ | $\mathbf{92.89 \pm 1.13}$ | $\mathbf{78.29 \pm 0.50}$ | $83.39 \pm 0.61$ |

Table 6: Acc comparison between global and per-module $\alpha$ strategies on ImageNet-R.

| Method | 5 tasks | 10 tasks | 20 tasks |
|---|---|---|---|
| Per-module $\alpha$ | 80.53 | 76.82 | 74.68 |
| Global $\alpha$ (ours) | 80.88 | 78.48 | 74.13 |

maintains plasticity comparable to that of the standard task vector, as also observed in prior work [64], while significantly reducing forgetting and thus improving overall performance.

Then we analyze its behavior through loss landscape visualizations and directional perturbation experiments. Specifically, the merged model is computed as: $\hat{\theta}_t = \beta \cdot \hat{\theta}_{t-1} + \alpha \cdot \theta_t^*$, where $\alpha$ is derived from Eq. 8 and $\beta = 1 - \alpha$.

Figure 2 visualizes the average loss landscape on ImageNet-R (left) and CUB (right). we visualize merging after Tasks 4 and 7. Each subplot spans a 2D convex space formed by $\hat{\theta}_{t-1}$ and $\theta_t^*$, where the horizontal axis indicates the task-specific model weight $\beta$, and the vertical axis indicates the previous model weight $\alpha$. Each point corresponds to a merged model, colored by its average loss across all learned tasks. **M** refers to only Infer after Merging without applying Train with perturbation.

**Observation ②: Convex Combination lies in Low-Loss Regions.** Our method determines a coefficient $\alpha$ and computes the merged parameter as a convex combination. This combination represents a direct interpolation between the starting and ending points of task 4's training. As the loss contours in all eight plots reveal, the convex paths between $\hat{\theta}_3$ and $\theta_4^*$, $\hat{\theta}_6$ and $\theta_7^*$ consistently lie in a low-loss region. This supports our design choice of convex combination over other merging methods, such as task arithmetic [65], which may move the merged model into unstable regions.

**Observation ③: Optimal Coefficients Locate Low-Loss Interpolation Points.** We next evaluate the effectiveness of the coefficients computed by our method. In each left-hand plot (without parameter perturbation), the merged point obtained using our optimal $\alpha$ consistently lies in a lower-loss region compared to either endpoint $\hat{\theta}_{t-1}$ or $\theta_t^*$. This indicates that our closed-form solution reliably identifies favorable interpolation points on the loss surface, providing empirical support for the theoretical foundation of our merging formulation.

**Observation ④: Task-Vector Perturbation Encourages Flat Minima and Better Generalization.** In each dataset's visualization, the left plot corresponds to training with standard cross-entropy (CE) loss, while the right plot depicts training with CE loss combined with parameter perturbation. The loss contours indicate that the additional perturbation enlarges the flatness and width of the low-loss basin around the merged model, making it more likely for the model to fall near an optimal region. This suggests that parameter perturbation helps avoid sharp minima and reduces parameter interference during model merging, thereby enhancing the generalization ability of the merged model. To further evaluate the importance of perturbation direction, we compare task-vector-based perturbation with random Gaussian noise that has the same Frobenius norm. As shown in Tab. 7, models perturbed along the task vector consistently outperform those using random noise. This demonstrates that applying perturbation in the direction of the task vector provides a reliable approximation to Eq. 11, thereby improving the performance of model merging.

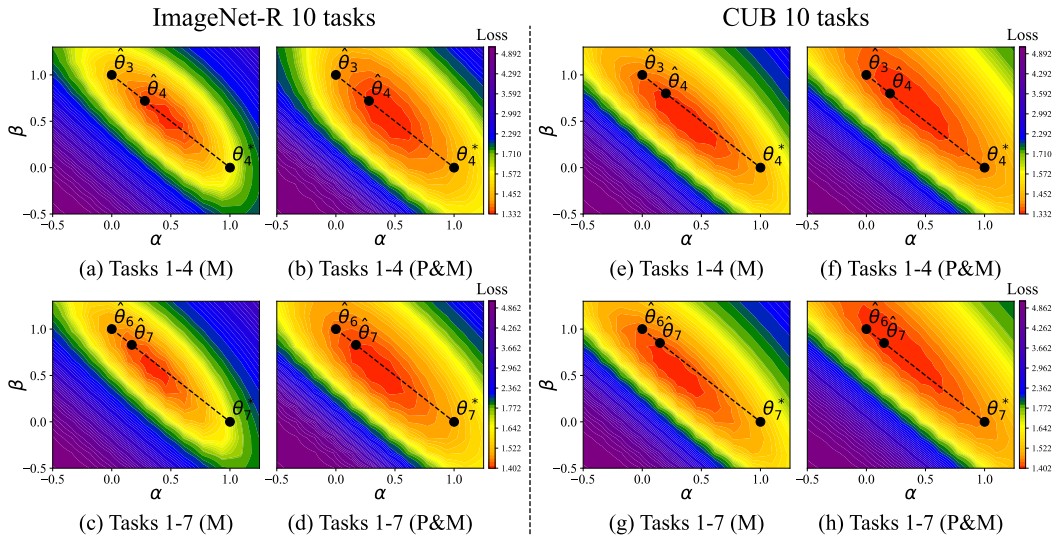

Figure 2: **Loss landscape visualization on ImageNet-R and CUB. M** denotes using only Infer after Merging (no perturbation during training). Each subplot shows the average loss surface after merging at Task 4 and Task 7, with axes representing weights $\alpha$ and $\beta$ in the convex combination $\hat{\theta}_t = \beta\hat{\theta}_{t-1} + \alpha\theta_t^*$. The convex path lies in a low-loss region (Obs. ②), and our optimal $\alpha$ consistently locates near the minimum (Obs. ③). Task-vector perturbation further enlarges the flat region (Obs. ④).

Table 7: The ablation study of the proposed P&M.

| Method | INR-10 | INR-20 | INA-10 | DN*-5 | C100-10 | CUB-10 |
|---|---|---|---|---|---|---|
| | Acc ↑ | Acc ↑ | Acc ↑ | Acc ↑ | Acc ↑ | Acc ↑ |
| LoRA | 65.72 | 56.35 | 44.41 | 71.81 | 72.58 | 64.82 |
| LoRA-M | 78.35 | 74.26 | 56.16 | 81.28 | 86.57 | 74.98 |
| LoRA-M w/ gauss noise | 78.48 | 74.13 | 49.51 | 83.00 | 85.83 | 74.09 |
| LoRA-P&M | **79.95** | **76.37** | **56.57** | **84.71** | **88.45** | **78.29** |

In summary, P&M improves CL through three key aspects: (1) Post-training scaling mitigates forgetting; (2) The convex combination of models, along with a theoretically grounded optimal coefficient, ensures that the merged model lies near an optimal region; (3) Task-vector perturbation perturbations enhance generalization and reduce parameter interference.

## 5 Conclusion

We propose P&M, a novel CL framework that incorporates post-training model merging into the learning paradigm. By combining a theoretically grounded merging strategy with task-vector-aligned perturbations, P&M effectively mitigates catastrophic forgetting. Our approach merges models via a convex combination of the current task optimum and the previous model, with the optimal coefficient derived from a closed-form, loss-based objective. To further enhance robustness, we introduce a lightweight regularization mechanism during training that applies stochastic perturbations along the task vector direction to improve the performance of the merged model. Integrated with LoRA, P&M offers a memory-efficient solution and achieves strong performance across various CL benchmarks. Experimental results show that unifying training-time dynamics with post-training merging provides a simple yet effective strategy for building continual learners with strong generalization and stability.

**Limitation.** Our method estimates the optimal merging coefficient using an analytical form based on the diagonal empirical Fisher Information Matrix. However, this diagonal approximation may not fully capture the true curvature of the loss landscape, and thus does not always guarantee optimality. Exploring more accurate yet efficient curvature approximations, is a direction for future work.

## Acknowledgment

Miao Zhang was partially sponsored by the National Natural Science Foundation of China under Grant 62306084 and U23B2051, Shenzhen College Stability Support Plan under Grant GXWD20231128102243003, and Shenzhen Science and Technology Program under Grant ZDSYS20230626091203008 and KJZD20230923115113026. Ziyue Qiao, School of Computing and Information Technology, Great Bay University, was supported by the National Natural Science Foundation of China (Grant No. 62406056) and the Guangdong Basic and Applied Basic Research Foundation (Grant No. 2024A1515140114).

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

# A Proof and Theoretical Details

## A.1 Proof of Eq. 8

To determine the optimal $\alpha_t^*$, we consider the following optimization problem:

$$\alpha_t^* = \arg\min_{\alpha_t} \sum_{i=1}^{t} \delta_i \approx \arg\min_{\alpha_t} \sum_{i=1}^{t} \frac{1}{2} \left(\hat{\theta}_{t-1} - \theta_i^*\right)^{\top} \mathbf{H}_i(\theta_i^*) \left(\hat{\theta}_{t-1} - \theta_i^*\right), \tag{18}$$

where $\hat{\theta}_{t-1}$ denotes the inference-time model parameters after learning on tasks 1 through $t-1$, and $\theta_i^*$ represents the optimal parameters for task $i$. The matrix $\mathbf{H}_i(\theta_i^*)$ denotes the Hessian of the loss function for task $i$ evaluated at its optimum $\theta_i^*$. Substituting $\hat{\theta}_t = \hat{\theta}_{t-1} + \alpha_t \Delta\theta_t^*$, into the objective, we obtain:

$$\sum_{i=1}^{t} \frac{1}{2} \left(\hat{\theta}_{t-1} + \alpha_t \Delta\theta_t^* - \theta_i^*\right)^{\top} \mathbf{H}_i(\theta_i^*) \left(\hat{\theta}_{t-1} + \alpha_t \Delta\theta_t^* - \theta_i^*\right). \tag{19}$$

Expanding the quadratic form inside the summation yields:

$$\sum_{i=1}^{t} \frac{1}{2} \left[(\hat{\theta}_{t-1} - \theta_i^*)^{\top} \mathbf{H}_i(\theta_i^*)(\hat{\theta}_{t-1} - \theta_i^*) + 2\alpha_t(\hat{\theta}_{t-1} - \theta_i^*)^{\top} \mathbf{H}_i(\theta_i^*) \Delta\theta_t^* + \alpha_t^2 (\Delta\theta_t^*)^{\top} \mathbf{H}_i(\theta_i^*) \Delta\theta_t^* \right]. \tag{20}$$

Define the objective function $J(\alpha_t)$ as:

$$J(\alpha_t) = \sum_{i=1}^{t} \frac{1}{2} \left[(\hat{\theta}_{t-1} - \theta_i^*)^{\top} \mathbf{H}_i(\theta_i^*)(\hat{\theta}_{t-1} - \theta_i^*) \right. \tag{21}$$

$$\left. + 2\alpha_t(\hat{\theta}_{t-1} - \theta_i^*)^{\top} \mathbf{H}_i(\theta_i^*) \Delta\theta_t^* + \alpha_t^2 (\Delta\theta_t^*)^{\top} \mathbf{H}_i(\theta_i^*) \Delta\theta_t^* \right]. \tag{22}$$

Taking the derivative of $J(\alpha_t)$ with respect to $\alpha_t$, we get:

$$\frac{dJ}{d\alpha_t} = \sum_{i=1}^{t} \left[(\hat{\theta}_{t-1} - \theta_i^*)^{\top} \mathbf{H}_i(\theta_i^*) \Delta\theta_t^* + \alpha_t (\Delta\theta_t^*)^{\top} \mathbf{H}_i(\theta_i^*) \Delta\theta_t^* \right]. \tag{23}$$

Setting $\frac{dJ}{d\alpha_t} = 0$ leads to the first-order optimality condition:

$$\sum_{i=1}^{t} (\hat{\theta}_{t-1} - \theta_i^*)^{\top} \mathbf{H}_i(\theta_i^*) \Delta\theta_t^* + \alpha_t \sum_{i=1}^{t} (\Delta\theta_t^*)^{\top} \mathbf{H}_i(\theta_i^*) \Delta\theta_t^* = 0. \tag{24}$$

Solving for $\alpha_t$, we obtain the optimal $\alpha_t$:

$$\alpha_t^* = -\frac{\sum_{i=1}^{t} (\hat{\theta}_{t-1} - \theta_i^*)^{\top} \mathbf{H}_i(\theta_i^*) \Delta\theta_t^*}{\sum_{i=1}^{t} (\Delta\theta_t^*)^{\top} \mathbf{H}_i(\theta_i^*) \Delta\theta_t^*}. \tag{25}$$

## A.2 Details of Eq. 11

Further, we want to reduce $\sum_{i=1}^{t} \delta_i$ to make the combined model better. It is evident that $\sum_{i=1}^{t} \delta_i \geq \sum_{i=1}^{t} \delta_i(\alpha_t^*)$ holds. We also have

$$\sum_{i=1}^{t} \delta_i(\alpha_t^*) = \frac{1}{2} \sum_{i=1}^{t} \left(\hat{\theta}_{t-1} - \theta_i^*\right)^{\top} \mathbf{H}_i \left(\hat{\theta}_{t-1} - \theta_i^*\right) - \frac{\left(\sum_{i=1}^{t} \left(\hat{\theta}_{t-1} - \theta_i^*\right)^{\top} \mathbf{H}_i \Delta\theta_t^*\right)^2}{2 \sum_{i=1}^{t} \Delta\theta_t^{*\top} \mathbf{H}_i \Delta\theta_t^*} \tag{26}$$

$$\leq \frac{1}{2} \sum_{i=1}^{t} \left(\hat{\theta}_{t-1} - \theta_i^*\right)^{\top} \mathbf{H}_i \left(\hat{\theta}_{t-1} - \theta_i^*\right) \tag{27}$$

The inequality holds because the second term is negative. We next explain why it is reasonable to ignore the second term and focus only on the first. The first term on the right-hand side dominates, as it is typically larger in magnitude and easier to compute, while the second term involves nested quadratic forms and is costly to evaluate. Therefore, we retain only the leading term as an approximation and obtain the upper bound:

$$\sum_{i=1}^{t} \delta_i(\alpha_t^*) \le \frac{1}{2} \sum_{i=1}^{t} \left( \hat{\theta}_{t-1} - \theta_i^* \right)^\top \mathbf{H}_i \left( \hat{\theta}_{t-1} - \theta_i^* \right) \tag{28}$$

We next show why the first term is the dominant one. We aim to show that for the decomposition:

$$\sum_{i=1}^{t} \delta_i(\alpha_t^*) = \frac{1}{2} \sum_{i=1}^{t} \left( \hat{\theta}_{t-1} - \theta_i^* \right)^\top \mathbf{H}_i \left( \hat{\theta}_{t-1} - \theta_i^* \right) - \frac{\left( \sum_{i=1}^{t} \left( \hat{\theta}_{t-1} - \theta_i^* \right)^\top \mathbf{H}_i \Delta\theta_t^* \right)^2}{2 \sum_{i=1}^{t} \Delta\theta_t^{*\top} \mathbf{H}_i \Delta\theta_t^*},$$

the second term is always less than or equal to the first, i.e.,

$$\frac{\left( \sum_{i=1}^{t} a_i^\top H_i v \right)^2}{\sum_{i=1}^{t} v^\top H_i v} \le \sum_{i=1}^{t} a_i^\top H_i a_i. \tag{29}$$

**Proof.** Let $A = \sum_{i=1}^{t} a_i^\top H_i a_i$ and $B = \sum_{i=1}^{t} v^\top H_i v$, and define:

$$C = \sum_{i=1}^{t} a_i^\top H_i v = \langle a, v \rangle_{\mathbf{H}}, \tag{30}$$

where $\langle \cdot, \cdot \rangle_{\mathbf{H}}$ denotes a generalized inner product defined via weighted Hessians.

Then by the Cauchy-Schwarz inequality in inner product spaces:

$$C^2 = \left( \sum_{i=1}^{t} a_i^\top H_i v \right)^2 \le \left( \sum_{i=1}^{t} a_i^\top H_i a_i \right) \cdot \left( \sum_{i=1}^{t} v^\top H_i v \right) = A \cdot B. \tag{31}$$

Therefore:

$$\frac{C^2}{B} \le A \quad \Rightarrow \quad \text{Second term} \le \text{First term.} \tag{32}$$

This proves that, in the decomposition where the objective is the first term minus the second, the second term is always less than or equal to the first. As a result, the first term dominates. Therefore, we focus on optimizing the first term through regularization, which is likely to reduce the overall objective as well.

## B  More Experimental Results and Details

### B.1  Details of Datasets

ImageNet-R contains 200 ImageNet [66] classes rendered in various artistic styles. ImageNet-A comprises 200 classes featuring natural adversarial examples that are typically misclassified by standard ImageNet-trained models. DomainNet covers $345$ object categories across six distinct visual domains. CIFAR100 is a classic image classification dataset with $60,000$ images equally distributed across 100 classes. CUB200 is a fine-grained bird classification dataset containing $11,788$ images over 200 categories.

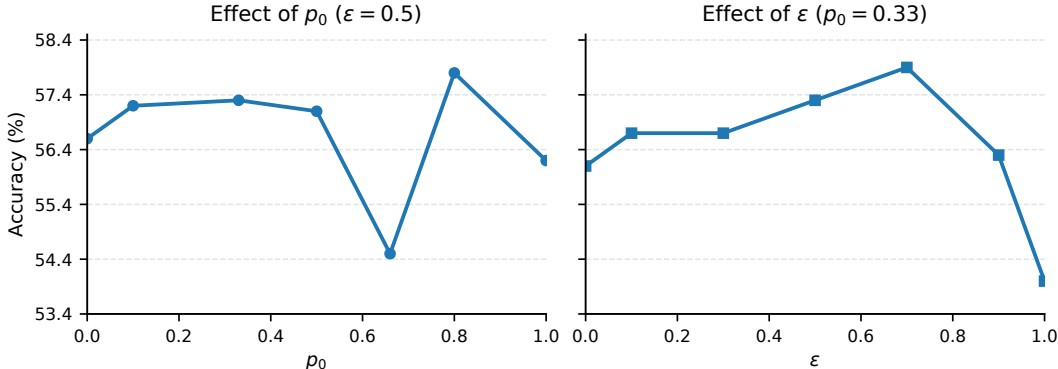

Figure 3: **Hyperparameter ablation on ImageNet-A.** Left: performance under varying $p_0$ values (probability of zero perturbation). Right: performance under different $\epsilon$ values (magnitude of perturbation).

## B.2 Hyperparameter ablation

We conduct ablation studies on the hyperparameters $p_0$ (probability of zero perturbation) and $\epsilon$ (perturbation magnitude) on the ImageNet-A dataset. As shown in Figure 3, we observe that setting $p_0 = 0.33$ and $\epsilon = 0.5$ yields consistently good performance, though not necessarily optimal. To reduce tuning overhead and maintain consistency, we use $p_0 = 0.33$ and $\epsilon = 0.5$ as default values for all experiments. Interestingly, when $p_0 = 0.33$, $p_0 = p_+ = p_-$.

## B.3 Memory and Training Time Analysis

We compare the memory footprint and training time of LORA and our proposed LORA-P&M on ImageNet-R and CIFAR100. As shown in Tab. 8, LORA-P&M introduces only a modest increase in memory usage—approximately 1.5 GB compared to standard LoRA—due to compute and store the diagonal empirical Fisher Information Matrix, while the cost introduced by stochastic perturbation is negligible. . Similarly, the training time increases by 0.2 to 0.4 hours per dataset. Despite this overhead, LORA-P&M achieves substantial accuracy gains: +14.23% on ImageNet-R and +15.87% on CIFAR100. These results demonstrate that our method offers a favorable trade-off between performance and resource efficiency.

Table 8: Comparison of memory usage, training time, and accuracy between model merging methods on ImageNet-R and CIFAR100.

| Method | ImageNet-R 10 tasks | | | CIFAR100 10 tasks | | |
|---|---|---|---|---|---|---|
| | Memory $\downarrow$ | Time $\downarrow$ | Acc $\uparrow$ | Memory $\downarrow$ | Time $\downarrow$ | Acc $\uparrow$ |
| LoRA | 22.80GB | 1.6h | 65.72 | 22.80GB | 3.1h | 72.58 |
| LoRA-P&M | 24.29GB | 1.8h | 79.95 | 24.29GB | 3.5h | 88.45 |

## B.4 Comparison with Recent Continual Learning Methods

To further contextualize our approach, we compare against several recent and competitive continual learning methods, including O-LoRA [67], Prompt Gradient Projection (PGP) [68], and Consistent Prompting (C-Prompt) [69].

Our method consistently outperforms prior approaches across all datasets and task splits, demonstrating strong adaptability and effective knowledge retention under the rehearsal-free continual learning setting.

Table 9: Comparison with recent continual learning methods. Results are reported as Acc on three standard benchmarks.

| Method | ImageNet-R 10 tasks | CIFAR100 10 tasks | CIFAR100 20 tasks |
|--------|--------|--------|--------|
| O-LoRA | 79.15 | 85.69 | 83.22 |
| DualP-PGP | 69.34 | 86.92 | 83.74 |
| C-Prompt | 77.14 | 87.82 | 83.97 |
| LoRA-P&M | **81.47** | **88.45** | **85.45** |

## B.5 Memory Efficiency and Online Fisher Approximation

One potential concern of our approach is that storing one Fisher information matrix per task leads to linearly increasing memory usage with the number of tasks. We further introduce an online variant of our method by maintaining a *running average* of the Fisher information across past tasks, resulting in a fixed-size Fisher representation. We denote this variant as LoRA-online P&M, and compare it with our standard LoRA-P&M framework.

Table 10: Comparison between LoRA-P&M and its online variant on ImageNet-R with different task splits. Values denote Acc (%).

| Method | ImageNet-R 5 tasks | ImageNet-R 10 tasks | ImageNet-R 20 tasks |
|--------|--------|--------|--------|
| LoRA-P&M | 81.47 | 79.95 | 76.37 |
| LoRA-online P&M | 79.97 | 78.12 | 71.20 |

As shown in Table 10, the online variant performs comparably to the original LoRA-P&M in the 5-task and 10-task settings. However, as the task sequence becomes longer (e.g., 20 tasks), the performance gradually degrades due to accumulated approximation error. Mitigating this trade-off between memory efficiency and performance stability is an important direction for our future work.

## B.6 Comparison with EWC

To further validate our approach, we additionally include experimental comparisons with **Elastic Weight Consolidation (EWC)** [70]. Since our method also leverages Fisher-based curvature approximations, EWC serves as a meaningful baseline. As shown in Table 11, our LoRA-P&M framework significantly outperforms LoRA-EWC across all benchmark settings.

Table 11: Comparison between LoRA-EWC and our proposed LoRA-P&M on ImageNet-R under different task splits. Results are reported as Acc (%).

| Method | ImageNet-R 5 tasks | ImageNet-R 10 tasks | ImageNet-R 20 tasks |
|--------|--------|--------|--------|
| LoRA-EWC | 71.47 | 65.42 | 55.67 |
| LoRA-P&M | **81.47** | **79.95** | **76.37** |

Although both EWC and our framework utilize the Fisher Information Matrix, the conceptual foundations and objectives of the two methods differ substantially. Our approach introduces a *model merging*-based inference paradigm, where a closed-form merging coefficient $\alpha_t^*$ is derived to minimize the total loss increase across all tasks. In contrast, EWC introduces a Fisher-weighted regularization term during training to penalize deviations from parameters that were important in previous tasks.

While EWC uses the Fisher matrix to estimate the *importance of each parameter* during sequential training, our method treats the Fisher matrix as a tractable approximation of the Hessian, which facilitates computing an optimal merging direction in the parameter space. Therefore, although we share the use of Fisher-based curvature information, our *formulation, objective, and application context* are fundamentally different.

**Algorithm 2** Diagonal Empirical Fisher

---

**Require:** Dataset $\mathcal{D} = \{(x_i, y_i)\}_{i=1}^N$, model $f_\theta$, loss $\ell(\cdot, \cdot)$, parameter-selection mask $m \in \{0, 1\}^d$
    (e.g., LoRA modules), optional mini-batch size $B$.

**Ensure:** Diagonal empirical Fisher $\widehat{F} \in \mathbb{R}_{\geq 0}^d$.

1: Initialize $\widehat{F} \leftarrow \mathbf{0}_d$
2: **for** each mini-batch $\mathcal{B} \subset \mathcal{D}$ of size $|\mathcal{B}| = B$ **do**
3:     $\mathcal{L}_\mathcal{B} \leftarrow \frac{1}{B} \sum\limits_{(x,y) \in \mathcal{B}} \ell\big(y, f_\theta(x)\big)$
4:     Compute gradient $g \leftarrow \nabla_\theta \mathcal{L}_\mathcal{B}$                                $\triangleright g \in \mathbb{R}^d$
5:     $g \leftarrow g \odot m$                $\triangleright$ keep only selected parameters (e.g., LoRA), zero otherwise
6:     $\widehat{F} \leftarrow \widehat{F} + g \odot g$                          $\triangleright$ element-wise square-and-sum
7: **end for**
8: $\widehat{F} \leftarrow \frac{1}{N} \widehat{F}$                             $\triangleright$ normalize by the number of samples
9: **return** $\widehat{F}$

---

