# OpenReview forum: "Train with Perturbation, Infer after Merging: A Two-Stage Framework for Continual Learning"
_NeurIPS.cc/2025/Conference — NeurIPS 2025 poster_

### Official Review · Reviewer_VKzv · 2025-06-02

**Clarity:** 3
**Significance:** 3
**Originality:** 3
**Rating:** 4
**Confidence:** 4

**Summary:**

The authors integrate continual learning with model merging to develop a single model capable of performing well across multiple tasks while mitigating catastrophic forgetting. They propose a method called *Infer after Merging*, which involves scaling the task vector ∆θ\*\_t = θ\**t − θ̂*{t−1} to reduce interference with previously acquired knowledge, and derive a closed-form solution to minimize overall performance degradation. Additionally, they introduce *Train with Perturbation*, which incorporates a regularization loss term based on the task vector and the Hessian matrix during training. Finally, they combine the P\&M approach with LoRA to reduce memory overhead.

**Questions:**

1. In line 149, under the maximum likelihood estimation setting, shouldn't the expected Fisher information matrix be equal to the negative of the Hessian matrix?
2. In Figure 2, the average loss landscape is visualized. It is unclear how this figure was constructed—what exactly does the "average" on the loss axis represent?
3. In line 139, should θ*_t be written as θ*_i to maintain consistency with the notation?

**Ethical Concerns:**

["NO or VERY MINOR ethics concerns only"]

**Final Justification:**

The main issue of this paper was the writing quality, which the authors have improved based on my suggestions.

**Limitations:**

Please provide the conditions under which the approximations or include error bounds to justify their use in the derivation.

**Quality:**

2

**Strengths And Weaknesses:**

**Strengths**

1. The paper provides a clear and well-structured introduction, effectively outlining the differences and connections between continual learning and model merging, along with a clear motivation for the proposed approach.
2. Experimental results consistently outperform the baseline methods, demonstrating the effectiveness of the proposed techniques.
3. The use of loss landscape visualizations offers intuitive insights into model behavior and optimization dynamics.

**Weaknesses**

1. The methodological derivation relies on several approximations—such as Taylor expansion, Fisher matrix estimation, Hessian matrix approximation, and symmetric finite differences—without sufficient justification for their validity. The potential cumulative error raises concerns. Can the actual benefit of minimizing the upper bound be empirically verified?
2. The related work section is brief. Some early studies on continual learning and model merging are neither discussed nor compared experimentally, making it difficult to assess the true advantages of the proposed method.
3. Although the method reduces memory overhead using LoRA, the computation and storage of the Fisher matrix still require significant resources. Is the approach inherently limited to LoRA-based fine-tuning?

---

> ### Author Rebuttal · Authors · 2025-07-31
>
> We sincerely thank the reviewer for their thoughtful comments and careful reading. Below we respond to each concern in detail.
>
> ---
>
> > W1:  Whether the theoretical approximations—such as Taylor expansion, Fisher approximation, and symmetric finite differences—are valid, and whether minimizing the upper bound brings empirical benefits.
>
>
> 1. We would like to clarify that the assumptions we adopt—such as approximating the Hessian with the empirical Fisher Information Matrix, applying a second-order Taylor expansion of the loss, and neglecting higher-order terms—are not unique to our work. These are widely used and well-accepted approximations in the continual learning [1][2][3].
>
> Such approximations have been extensively validated and empirically supported in recent deep learning theory and algorithms. Our use of them follows established practices rather than introducing new or unverified assumptions.
>
> 2. To empirically evaluate the benefit of minimizing the upper bound (Eq. 11), we conducted ablation experiments:
>
> - LoRA-M (with merging but no perturbation) vs LoRA-P&M (with perturbation): The latter consistently improves performance, as shown in Table 4 and Figure 1.
> - Additionally, in Figure 2, we visualize the loss surface across merged models. The optimal α derived from the upper-bound formulation consistently lands in low-loss regions.
> - We also show in table below that adding our perturbation to external merging methods (e.g., CoFiMA(CoFiMA-P)) improves performance—indicating that minimizing the derived upper bound brings empirical benefits beyond our method.
>
> | Method       | ImageNet-R 5tasks | ImageNet-R 10tasks  | ImageNet-R 20tasks  |
> |--------------|----------|--------|-------------|
> |LoRA-CoFiMA       | 80.26   | 79.06 | 75.09    |
> | LoRA-CoFiMA-P     | **81.03**    | **79.70**  |**76.17**       |
>
> While the approximations are imperfect, the **combined analytical formulation and empirical outcomes** strongly suggest that they are effective and valid in practice.
>
> ---
>
> > W2: The related work section is brief. Some early studies on continual learning and model merging are neither discussed nor compared experimentally.
>
>
> Thank you for pointing this out. We agree that the related work section can be expanded. In our revised version, we will add discussion of classical continual learning methods such as EWC, GEM, and  cite early model merging works then discuss distinctions from our coefficient-guided merging.
>
> Furthermore, we have included experimental comparisons with **EWC** in our updated results. The results show that LoRA-P&M substantially outperforms EWC.
>
> | Method       | ImageNet-R 5tasks | ImageNet-R 10tasks  | ImageNet-R 20tasks  |
> |--------------|----------|--------|-------------|
> |LoRA-EWC      | 71.46  | 65.42 | 55.67  |
> | LoRA-P&M     | **81.47**   |**79.95** | **76.37**     |
>
> We also conducted additional comparisons with several recent continual learning methods:
> | Method       | ImageNet-R 10tasks | CIFAR100 10tasks | CIFAR100 20tasks |
> |--------------|----------|--------|-------------|
> | OLoRA[4]      | 79.15  | 85.69 | 83.22     |
> | DualP-PGP[5]     | 69.34    | 86.92  |83.74       |
> | CPrompt[6]      | 77.14   | 87.82  | 83.97       |
> | LoRA-P&M     | 81.47   | 88.45  | 85.45      |
>
>
> ---
>
>
> > W3: The computation and storage of the Fisher matrix require significant resources. Is the approach inherently limited to LoRA-based fine-tuning?
>
> We appreciate the reviewer’s concern and would like to clarify that while our method is currently implemented with LoRA-based fine-tuning, it is not inherently restricted to LoRA. The core of our approach—the task vector merging strategy and curvature-guided perturbation—can be applied to more PEFT methods, including but not limited to adapters, prompt tuning.
>
> In our setup, the actual overhead is modest for the following reasons:
>
> We only compute and store the diagonal Fisher matrix for modules equipped with LoRA, which, in our implementation, are limited to the key and value projections of each attention module. These parameters account for approximately one-sixth of the total model parameters. Moreover, in practice, since the diagonal Fisher matrix is only used at the end of training for each task, it can be stored on the CPU or disk, preventing linear growth in GPU memory usage.
>
> As shown in Table 7, the overall memory increase from LoRA(22.80GB) to LoRA-P&M(24.29GB) is approximately 1.5 GB for a dataset with 10 tasks.
>
> While Section 3.4 introduces LoRA primarily as a means to reduce memory overhead, we emphasize that LoRA is also better aligned with the goals of continual learning (CL) beyond efficiency. LoRA tends to cause less interference with previously learned tasks[1], allows for easier task-specific adaptation, and requires fewer training examples and parameters to achieve strong performance. In contrast, full fine-tuning often leads to greater forgetting and lacks modularity. Therefore, even when memory is not a limiting factor, LoRA remains a more suitable and effective paradigm for continual learning.
>
>
>
>
> ---
>
> > Q1&Q3: Typos
>
> Thank you for this sharp observation.  We will correct this in the revision.
>
> ---
>
> > Q2: In Figure 2, the average loss landscape is visualized. It is unclear how this figure was constructed—what exactly does the "average" on the loss axis represent?
>
> Thank you for pointing this out. In Figure 2:
>
> - The “average loss” refers to the **mean task-wise cross-entropy loss** computed on validation data from all tasks seen up to that point.
> - Each point on the 2D plane corresponds to a convex combination of $\hat{\theta}_{t-1}$ and $\theta^*_t$, i.e., $\theta = \beta \hat{\theta}_{t-1} + \alpha \theta^*_t$.
> - We evaluate the merged model at that point and compute its average loss across all past tasks.
>
> We will clarify this in the figure caption and the main text.
>
> ---
>
> We appreciate the reviewer’s close reading and constructive feedback. The points raised will be addressed thoroughly in the revised version, and we thank you again for helping us improve the clarity and rigor of the work.
>
>
> [1] James Kirkpatrick, et al. "Overcoming Catastrophic Forgetting in Neural Networks." 2017.
>
> [2] Zhenyi Wang, et al. "A UNIFIED AND GENERAL FRAMEWORK FOR CONTINUAL LEARNING." ICLR24.
>
> [3] Mei Li, et al. "BECAME: BayEsian Continual Learning with Adaptive Model MErging." ICML25.
>
> [4] Xiao Wang, et al. "Orthogonal Subspace Learning for Language Model Continual Learning."  EMNLP23.
>
> [5] Jingyang Qiao, et al. "Prompt Gradient Projection for Continual Learning."  ICLR24.
>
> [6] Zhanxin Gao, et al. "Consistent Prompting for Rehearsal-Free Continual Learning." CVPR24
>
> [7] Dan Biderman, et al. "LoRA Learns Less and Forgets Less." TMLR24.

---

> > ### Comment · Reviewer_VKzv · 2025-08-06
> >
> > Thank you for your response and clarifications. I appreciate the additional experiments.
> >
> > One suggestion for further improvement is to enhance the transparency and reproducibility of the experimental setup. For example, it would be helpful if the authors could provide more details regarding the hyperparameter settings, data preprocessing steps, and the exact protocol for computing the Fisher matrix.

---

> > > ### Author Response · Authors · 2025-08-06
> > >
> > > We sincerely thank the reviewer for the constructive suggestion regarding transparency and reproducibility.
> > >
> > >
> > > The experimental setup for our proposed method is described in detail in Section 4.1 of the manuscript. For the compared baselines, we follow their default hyperparameter settings. For example, in CoFiMA, the hyperparameter $\lambda$ is set to 0.4 as in the original paper. The data preprocessing steps and the procedure for computing the Fisher matrix can be found in our open-source code repository.
> > >
> > > To further improve transparency and reproducibility, we will provide detailed descriptions of the above aspects in the appendix of the revised version as well as in the README of the open-source code. For example, we will add the algorithmic procedure for computing the **diagonal empirical Fisher information matrix** as follows:
> > >
> > > 1. Initialize $\mathbf{\hat F} \leftarrow \mathbf{0}_d$
> > > 2. **for** each $(x_i, y_i) \in \mathcal{D}$ **do**
> > >    - Forward pass: $\hat y_i \leftarrow f_\theta(x_i)$
> > >    - Compute loss and Backward pass: $g_i \leftarrow  \nabla_\theta \mathcal{L}(y_i, \hat{y}_i)$
> > >    - Update: $\mathbf{\hat F} \leftarrow \mathbf{\hat F} + g_i \odot g_i$  // Element-wise square and sum
> > > 3. Normalize: $\mathbf{\hat F} \leftarrow \frac{1}{N} \mathbf{\hat F}$  // Average over all $N$ samples
> > >
> > > We hope that our clarifications and additional experiments have addressed the concerns raised. We believe these improvements significantly strengthen the paper, and we kindly invite the reviewer to consider this when assessing the final score.

---

### Official Review · Reviewer_47Vz · 2025-06-30

**Clarity:** 2
**Significance:** 3
**Originality:** 3
**Rating:** 4
**Confidence:** 4

**Summary:**

The paper proposes a model merging idea for continual learning. The idea to continually train from a base model for a sequence of tasks and take a snapshot of the weights after each task is trained. And at inference, the weight combination of the weight snapshots is used. A simple idea that seems effective according to the experiments.

**Questions:**

see above

**Ethical Concerns:**

["NO or VERY MINOR ethics concerns only"]

**Final Justification:**

Authors' response has addressed all of my concerns, and I'm increasing the score to 4.

**Limitations:**

yes

**Quality:**

3

**Strengths And Weaknesses:**

## Strengths
1. The idea of merging task snapshots makes sense, and the EWC-inspired regularisation is intuitive.
2. The merging coefficient is obtained using a Fisher-based curvature approximation -- previously used in continual learning.
3. Results show consistent improvements with the proposed method, compared to other fine-tuning or prompt-based methods.

## Weaknesses
1. Previous continual learning methods are not compared. These methods were developed before the introduction of foundational models; however, they can be easily incorporated into the pretraining+finetuning framework. Many notable continual methods include EWC, GEM [a] and their variants. Given the similarity to EWC, these methods need to be compared.
2. Related to the above point, the perturbation-based method and the merging coefficient derivation have similarities with EWC and should be credited, in my opinion.
3. The weight coefficient is mentioned as "optimal" in the abstract; however, it is not optimal, and it is derived for a Fisher-based second-order approximation of the loss.
4. How does the order of tasks affect the performance? If the weights are sequentially trained, it seems that the task order would matter, and it can have an impact on the final performance.
- Why not train for all the tasks from the base model and do the merging? How would this compare to the proposed method?
5. The memory grows linearly with the no of tasks, anyway to reduce this?
6. The experiment setup is not clear. Is the task ID known at inference (multi-head or single-head evaluation) [b]?

## References
- [a] Lopez-Paz, David, and Marc'Aurelio Ranzato. "Gradient episodic memory for continual learning." Advances in neural information processing systems 30 (2017).
- [b] Chaudhry, Arslan, et al. "Riemannian walk for incremental learning: Understanding forgetting and intransigence." Proceedings of the European conference on computer vision (ECCV). 2018.

---

> ### Author Rebuttal · Authors · 2025-07-31
>
> We sincerely thank the reviewer for their insightful and detailed feedback. Below we address each point individually.
>
> ---
>
> > W1: Comparison with methods include EWC and GEM.
>
>
> We have now included experimental comparisons with **EWC** in our updated evaluations. Since our method also leverages Fisher-based curvature approximation, we believe this is a meaningful baseline. The results show that LoRA-P&M significantly outperforms EWC across all benchmarks.
>
> | Method       | ImageNet-R 5tasks | ImageNet-R 10tasks  | ImageNet-R 20tasks  |
> |--------------|----------|--------|-------------|
> |LoRA-EWC      | 71.47  | 65.42 | 55.67  |
> | LoRA-P&M     | **81.47**   |**79.95** | **76.37**     |
>
> As for **GEM**, we respectfully did not include it because it is based on **experience replay**, which violates the **rehearsal-free continual learning assumption** adopted in our work. Our method, like SD-LoRA and InfLoRA, operates under a constraint that previous data cannot be revisited. Thus, GEM is not directly comparable in our setting.
>
> ---
>
> > W2: A detailed comparison with EWC.
>
>
> We would like to clarify that our two-stage framework is conceptually distinct from EWC. Our approach introduces a *model merging*-based inference paradigm, where a closed-form merging coefficient $\alpha_t^*$ is derived to minimize total loss increase across all tasks.
> In contrast, EWC adds a Fisher-weighted regularization term during training to penalize deviations from previously important parameters. In EWC, the Fisher matrix serves to estimate the *importance of each parameter* for past tasks. In our case, the Fisher matrix is used solely as a tractable approximation of the Hessian in order to compute an optimal merging direction. Thus, while we share a technical approximation with EWC, our theoretical formulation, implementation, and application context are entirely different.
>
> We appreciate the reviewer’s observation regarding the connection to EWC. While both our method and EWC utilize the Fisher Information Matrix, the motivation and role of this approximation are fundamentally different.
> We will clarify this distinction in the revised version and cite EWC appropriately.
>
> ---
>
> > W3: The expression of "optimal" in the paper is not sufficiently rigorous.
>
>
> Thank you for highlighting this. Our merging coefficient $\alpha_t^*$ is **theoretically optimal under the Fisher-based second-order loss approximation**, but we agree that this does not imply global optimality with respect to the original loss surface.
>
> To prevent misunderstanding, we will revise our language throughout the paper (including the abstract) to refer to α as **“analytically derived under a curvature approximation”**, and clarify its theoretical basis and limitations.
>
> ---
>
> > W4: To what extent does the task order influence the performance of our method? How would the results change if all sub-models were initialized from the same base model rather than sequentially?
>
>
> 1. Indeed, task order can influence performance in continual learning. In all of our experiments, **we train three runs under different random seeds**, each corresponding to **a different task order**. We report mean and standard deviation to capture this variability. We will clarify this in the experimental setup.
>
> 2. We agree this is an important baseline. In fact, Table 3 in our paper already includes such methods:
> - **Model Averaging**: directly merges task-specific models trained from the same base model.
> - **DARE**: a recent data-free merging method.
>
> ---
>
> > W5: The memory grows linearly with the no of tasks, anyway to reduce this?
>
> As you noted, storing one Fisher matrix per task results in linear memory growth. However:
> - We only store **diagonal Fisher approximations** for low-rank **LoRA modules**, making the per-task cost minimal (Table 7 shows ~1.5GB/24.29GB across 10 tasks). Moreover, in practice, since the diagonal Fisher matrix is only used at the end of training for each task, it can be stored on the CPU or disk, preventing linear growth in GPU memory usage.
> - Inspired by **Online EWC** [1], we plan to replace per-task Fisher storage with a simple **running average** over past tasks, resulting in a **fixed-size** Fisher representation. We denote this variant as *Online P&M*, and provide experimental comparisons against our original P&M framework.
>
>
>
> | Method       | ImageNet-R 5tasks | ImageNet-R 10tasks  | ImageNet-R 20tasks  |
> |--------------|----------|--------|-------------|
> |LoRA-P&M     | **81.47**   |**79.95** | **76.37**     |
> |LoRA-online P&M     | 79.97  | 78.12 | 71.2  |
>
> In the 5-task and 10-task settings, the approach of aggregating Fisher information from all previous tasks performs comparably to our LoRA-P&M method. However, when the task sequence becomes longer (e.g., 20 tasks), its performance degrades significantly. Addressing this limitation is one of our directions for future research.
>
>
> ---
>
> > W6: The experiment setup is not clear. Is the task ID known at inference (multi-head or single-head evaluation).
>
> We follow the **class-incremental learning (CIL)** setting, where the task ID is **not provided at inference time**, and every tasks has a classifier head. Each task expands the output space (e.g., from 20 classes to 200 classes across 10 tasks in CIFAR100), and the model must distinguish among **all seen classes**.
>
> This is considered the **most challenging and realistic** continual learning setting, and aligns with the evaluation protocols used in SD-LoRA, CODA-Prompt, and other recent works. We will clarify this explicitly in the revised experimental setup.
>
> ---
>
> We again thank the reviewer for their constructive suggestions. We will revise the manuscript to reflect the above clarifications, add missing citations, and expand our baseline coverage accordingly.
>
> [1] Ferenc Huszar, et al. "On Quadratic Penalties in Elastic Weight Consolidation." 2017.

---

> > ### Comment · Reviewer_47Vz · 2025-08-02
> > **Thanks for addressing the concerns**
> >
> > Thanks for addressing my concerns and providing comparisons with EWC. Please include them in the revised version along with other clarifications regarding the experiment setup.

---

> > > ### Author Response · Authors · 2025-08-02
> > >
> > > Dear Reviewer 47Vz,
> > >
> > > Your feedback has greatly helped us improve the clarity and completeness of the paper. Thank you very much!
> > >
> > > We will make sure to incorporate the key clarifications and experiment from this discussion into the revised manuscript.
> > >
> > > Best regards,
> > >
> > > Authors

---

### Official Review · Reviewer_AEtr · 2025-07-05

**Clarity:** 2
**Significance:** 3
**Originality:** 3
**Rating:** 4
**Confidence:** 4

**Summary:**

This paper presents LoRA-P&M, a continual learning (CL) method that applies model merging to LoRA components across tasks. To enable effective merging and improve generalization, two techniques are introduced: (1) a theoretically grounded merging coefficient, $\alpha$, computed using Fisher matrices; and (2) perturbations along task vectors introduced during training. Experimental results show that LoRA-P&M outperforms existing methods on multiple CL benchmarks. Loss landscape visualizations further suggest that it promotes flatter minima for the merged models.

**Questions:**

Besides the weakness shown in the above section, please also see the following questions:

Q1: Since perturbation training aims to optimize a more complex loss function (i.e., Equation 14), does it lead to slower convergence?

Q2: Given that perturbation training leads to flatter minima, as shown in Figure 2, could it also benefit other LoRA-based continual learning methods, such as SD-LoRA?

Q3: LoRA is introduced in Section 3.4 mainly to address memory consumption. If memory were not a limiting factor, would the proposed method still be effective—or perhaps perform even better?

**Ethical Concerns:**

["NO or VERY MINOR ethics concerns only"]

**Final Justification:**

The rebuttal and follow-up discussion with the authors have addressed the main concerns I raised in the initial review:

- Additional costs: The authors clarified the storage overhead associated with the Fisher matrix and presented experimental results demonstrating the method’s convergence speed, which alleviated my concerns regarding storage cost and training efficiency.

- Comparison with recent methods: The authors provided additional comparisons with recent approaches, including O-LoRA, PGP, and C-Prompt, and showed that the proposed method achieves better performance.

- Effectiveness relative to CoFiMA: The authors clarified the experimental results and included further comparisons to better demonstrate the effectiveness of the proposed method compared to CoFiMA.

Given that these key concerns have been addressed, I have updated my assessment. I intend to raise my score from 3 to 4 accordingly.

**Limitations:**

The paper acknowledges a limitation that the Fisher matrix may not fully capture the true curvature of the loss landscape. Another potential limitation is that storing a Fisher matrix for each task results in memory usage that grows linearly with the number of tasks, which may hinder scalability to large task sequences.

**Paper Formatting Concerns:**

I do not notice any formatting issues in the paper.

**Quality:**

2

**Strengths And Weaknesses:**

Strength:
- The core idea adopts a simple yet effective model merging strategy for continual learning, supported by additional techniques that enhance its performance.

- Tables 2, 3, 5, and 6 show promising results compared to LoRA-based and prompt-based continual learning methods across diverse datasets.

- The analysis in Figure 2 is insightful and highlights the benefits of perturbation training.

Weakness:
- LoRA-P&M requires storing a Fisher matrix $F_i$ for each task, leading to memory growth that is linear in the number of tasks.

- Although the paper claims to achieve state-of-the-art performance (Lines 20-21 in the abstract), several relevant continual learning methods are missing from the comparison. For example, recent LoRA-based approaches such as I-LoRA [1] and O-LoRA [2] are not included. Moreover, several newer prompt-based methods are also omitted [3][4][5][6].

- Equation 17 in Section 3.4 is somewhat unclear. Since LoRA parameters are typically inserted at different locations within the network, it is not immediately clear why they are summed in this equation.

- The idea of weighted model parameter combination (i.e., model merging) in continual learning has also been explored in CoFiMA [7]. However, when comparing the ablated version of LoRA-P&M (i.e., LoRA-M in Table 4) with CoFiMA (Table 3), LoRA-M appears to perform worse. This raises the question of whether the proposed weighted model merging strategy—specifically, the use of the optimal merging coefficient $\alpha$—is less effective than CoFiMA's approach.


[1] 2025 ICASSP Analyzing and Reducing Catastrophic Forgetting in Parameter Efficient Tuning.

[2] 2023 EMNLP Orthogonal Subspace Learning for Language Model Continual Learning.

[3] 2024 ECCV RCS-prompt: Learning Prompt to Rearrange Class Space for Prompt-based Continual Learning.

[4] 2024 ICLR Prompt Gradient Projection for Continual Learning.

[5] 2024 ECCV PromptFusion: Decoupling Stability and Plasticity for Continual Learning.

[6] 2024 CVPR Consistent Prompting for Rehearsal-Free Continual Learning.

[7] 2024 ECCV Weighted Ensemble Models Are Strong Continual Learners

---

> ### Author Rebuttal · Authors · 2025-07-31
>
> We sincerely thank the reviewer for their thoughtful and constructive feedback. Below we address each point in detail and clarify the concerns.
>
> ---
>
> > W1: LoRA-P&M requires storing a Fisher matrix for each task, leading to memory growth that is linear in the number of tasks.
>
> In our setup, the actual overhead is modest for the following reasons:
>
> We only compute and store the diagonal Fisher matrix for modules equipped with LoRA, which, in our implementation, are limited to the key and value projections of each attention module. These parameters account for approximately one-sixth of the total model parameters. Moreover, in practice, since the diagonal Fisher matrix is only used at the end of training for each task, it can be stored on the CPU or disk, preventing linear growth in GPU memory usage.
>
> As shown in Table 7, the overall memory increase from LoRA(22.80GB) to LoRA-P&M(24.29GB) is approximately 1.5 GB for a dataset with 10 tasks.
>
> ---
>
> > W2: Comparison with more continual learning methods
>
> We appreciate the reviewer pointing out recent methods. Due to time constraints, we were able to compare against **O-LoRA**, **Prompt Gradient Projection(PGP)**, and **Consistent Prompting(C-Prompt)**. These represent strong and recent methods across both LoRA-based and prompt-based paradigms.
>
> Regarding I-LoRA and PromptFusion: these methods rely on data replay, which fundamentally differs from our task setting (rehearsal-free continual learning). As such, we respectfully argue that they are not directly comparable to our approach.
>
> | Method       | ImageNet-R 10tasks | CIFAR100 10tasks | CIFAR100 20tasks |
> |--------------|----------|--------|-------------|
> | OLoRA      | 79.15  | 85.69 | 83.22     |
> | DualP-PGP     | 69.34    | 86.92  |83.74       |
> | CPrompt      | 77.14   | 87.82  | 83.97       |
> | LoRA-P&M     | 81.47   | 88.45  | 85.45      |
>
> We will add the comparisons and discussion with these methods to the revised version of our paper.
>
> ---
>
>
> > W3: Clarification of Equation 17
>
> Thank you for the helpful comment. We agree that the current form of Equation 17 may lead to confusion. Specifically, it may be misinterpreted as a global parameter-wise summation, whereas in practice, each LoRA update is applied **locally** and independently at specific network locations (e.g., attention key or value projections).
>
> To clarify this, we will revise Equation 17 as:
>
> $\theta_t = \theta_{t-1} + \bigoplus_{p \in l} A_t^{(p)} B_t^{(p)}$
>
> where $\bigoplus$ denotes *location-wise module insertion*, not element-wise addition. Each low-rank update $A_t^{(p)} B_t^{(p)}$ is injected into the pre-trained weight at its corresponding module $p$, and does not affect other components of the model.
>
> ---
>
> > W4: Detailed Comparison with CoFIMA
>
>
> We thank the reviewer for raising this important comparison. In table  below, we report the full LoRA-M versus CoFiMA across multiple benchmarks, showing that the performances are very close:
>
> | Method       | average FAA on 5 datasets|
> |--------------|----------|
> | LoRA-CoFiMA       | 75.52    |
> | LoRA-M     |  75.27   |
>
> To further analyze, we conducted a **CoFiMA + Perturbation** experiment (i.e., applying our task-vector perturbation during training and using CoFiMA merging). The results show improved performance over plain CoFiMA, but still underperform our full LoRA-P&M method. This indicates that:
> - Our second-stage theoretical derivation is based on the model merging perspective, suggesting that perturbation is generally beneficial for merging-based strategies. As shown in Figure 2, such perturbation indeed improves the flatness of the loss landscape around the merged model;
> - Our closed-form optimal merging coefficient and the perturbation are mutually reinforcing, yielding the best result.
>
> | Method       | ImageNet-R 5tasks | ImageNet-R 10tasks  | ImageNet-R 20tasks  |
> |--------------|----------|--------|-------------|
> |LoRA-CoFiMA       | 80.26   | 79.06 | 75.09    |
> | LoRA-CoFiMA-P     | **81.03**    | **79.70**  |**76.17**       |
> | LoRA-M       | 80.86     | 78.35  | 74.26        |
> | LoRA-P&M     | **81.47**   |**79.95** | **76.37**     |
>
> ---
>
> > Q1: Whether the perturbation may slow down the convergence speed
>
> We measured convergence speed in terms of loss decay per training step. As shown in the  table below as a example, perturbation training (LoRA-P&M) converges at a comparable rate to standard LoRA:
>
> | Train Loss of Task 8(ImageNet-R 10tasks)     | 1 % Steps  | 2 % Steps| 10 % Steps  |
> |--------------|-----------------------------|---------------|------
> | LoRA-M        | 1.90                       | 0.70         |0.26
> | LoRA-P&M     | 1.92                        |0.71         | 0.25
>
> The added regularization term leads to slightly more stable but not slower convergence, confirming the efficiency of the stochastic perturbation.
>
> ---
>
> > Q2: Whether the proposed perturbation strategy remains effective when integrated with other continual learning methods, such as SD-LoRA.
>
>
> We explored this question by applying our perturbation scheme to **SD-LoRA**, a state-of-the-art LoRA-based continual learning method. Interestingly, we observed **performance degradation** when adding perturbation to SD-LoRA:
>
> | Method       | ImageNet-R 5tasks | ImageNet-R 10tasks  | ImageNet-R 20tasks  |
> |--------------|----------|--------|-------------|
> | SD-LoRA        | 79.15   | 77.34 | 75.26    |
> | SD-LoRA + Perturbation    | 76.9    | 75.6  |72.22       |
>
>
> This suggests that our perturbation training is particularly effective for **merging-based strategies** like P&M or CoFiMA, where parameter interpolation plays a central role in preserving task knowledge.
>
> In contrast, SD-LoRA involves **dynamically optimizing task-specific vectors** during training. Our perturbations, which are aligned with a fixed task vector direction, may interfere with SD-LoRA’s learning dynamics and thus reduce performance.
>
> This further highlights that the benefit of perturbation is **targeted toward model merging**, and not necessarily applicable to all continual learning strategies.
>
> ---
>
> > Q3: Whether the proposed method would be more effective when applied on top of non-PEFT methods.
>
>
> While Section 3.4 introduces LoRA primarily as a means to reduce memory overhead, we emphasize that LoRA is also better aligned with the goals of continual learning (CL) beyond efficiency. LoRA tends to cause less interference with previously learned tasks[1], allows for easier task-specific adaptation, and requires fewer training examples and parameters to achieve strong performance. In contrast, full fine-tuning often leads to greater forgetting and lacks modularity. Therefore, even when memory is not a limiting factor, LoRA remains a more suitable and effective paradigm for continual learning.
>
>
>
> ---
>
> We again thank the reviewer for their helpful feedback. We will incorporate the clarifications, additional comparisons, and ablations in the revised version.
>
>
> [1] Dan Biderman, et al. "LoRA Learns Less and Forgets Less." TMLR24.

---

> > ### Comment · Reviewer_AEtr · 2025-08-06
> >
> > Thank you to the authors for the detailed responses and additional experiments. The rebuttal has addressed most of my concerns. I have just one remaining question regarding the response to “W4: Detailed Comparison with CoFIMA.”
> >
> > While I understand that perturbation training can improve the performance of another merging-based LoRA-CoFiMA, my original question was why the closed-form optimal merging coefficient $\alpha$ used in LoRA-M performs worse than the heuristic merging coefficient derived from the Fisher matrix in LoRA-CoFiMA. Could the authors elaborate on any potential factors that might explain this outcome?

---

> ### Author Response · Authors · 2025-08-06
>
> Dear Reviewer AEtr,
>
> Thank you for your follow‑up question and for the constructive feedback. Regarding your remaining question, we would like to provide the following clarification:
>
> Our merging method is derived from minimizing the total loss across all tasks, yielding a theoretically near‑optimal merging coefficient that requires **no additional hyperparameters**. Since computing the Hessian exactly is prohibitively expensive, we approximate it using the diagonal of the empirical Fisher information matrix, which inevitably introduces estimation errors [1]. This limitation has been discussed in the *Limitation* section of our paper. As a result, as shown in Fig. 2, the coefficient obtained by LoRA‑M via Eq. 8 does not always lie at the true loss minimum.
>
> In contrast, CoFiMA guides model merging by introducing a hyperparameter $\lambda$, which leaves room for tuning and **requires additional hyperparameter adjustment**. The ablation study in the Fig. 4 of  the original CoFiMA paper also shows that its performance is relatively sensitive to the hyperparameter \(\lambda\). Fig. 2 also shows that, due to the inherent robustness of the pretrained model, **a region of \(\alpha\) values yields similar loss**, so CoFiMA with a tunable hyperparameter can also achieve good performance. Moreover, as noted in our earlier rebuttal content, LoRA‑M demonstrates better compatibility with perturbation‑based training compared to CoFiMA.
>
> We thank you again for your valuable feedback, and we hope our explanation addresses your concern. We would be glad to discuss further if you have any additional questions.
>
> [1] Gido M. van de Ven. "On the Computation of the Fisher Information in Continual Learning." arXiv preprint arXiv:2502.11756.
>
> Best regards,
>
> Authors

---

> > ### Comment · Reviewer_AEtr · 2025-08-07
> >
> > Thank you for your clarifications. All of my concerns have been addressed, and I intend to raise my score accordingly.

---

> > > ### Author Response · Authors · 2025-08-07
> > >
> > > Dear Reviewer AEtr,
> > >
> > > Thank you very much for your kind follow-up and for deciding to raise your score.  We sincerely appreciate your thoughtful evaluation and are glad that our clarifications helped address your concerns.  Your recognition and support are truly encouraging to us.
> > >
> > > Best regards,
> > >
> > > Authors

---

### Official Review · Reviewer_Fr4E · 2025-07-06

**Clarity:** 4
**Significance:** 2
**Originality:** 3
**Rating:** 4
**Confidence:** 1

**Summary:**

This paper introduces the concept of model merging into continual learning. Unlike traditional continual learning approaches, the proposed P&M, after acquiring parameters for the new task, employs an parameter interpolation method to balance the model performance across both old and new tasks. The interpolation weights are theoretically derived. Furthermore, the authors incorporate regularization term during training to enhance the effectiveness of model merging.

**Questions:**

(1) Please provide more/additional justification for the necessity of the two-stage approach. For example, in what scenarios does this method demonstrate superiority in terms of performance, efficiency, or practicality? Including concrete examples would also help readers build an intuitive understanding of its benefits.

(2) Is it possible to add **α** during the derivation of training with perturbation?

(3) Can more parameter statistics be added to demonstrate the validity of the hypothesis proposed in the paper?

**Ethical Concerns:**

["NO or VERY MINOR ethics concerns only"]

**Final Justification:**

The main reason why I modified the score is that the authors explained the rationale for incorporating model merging into continuous learning. In addition, the authors conducted experiments to demonstrate that the performance of parameter-independent “α” (i.e. global “α”) is as competitive as learning separate “α” values for each LoRA module.

**Limitations:**

yes

**Quality:**

3

**Strengths And Weaknesses:**

Strengths

S1: This paper proposes a novel continual learning paradigm. The authors have conducted detailed theoretical derivations tailored to this new paradigm, along with reasonable training perturbation and model merging methods. The work shows high originality

S2: The derivations are rigorous and appear with no obvious major weaknesses. The conclusion seems sound and well-supported.

S3:  The writing is clear and well-structured. Experiments are conducted on multiple datasets, enhancing the reliability and credibility of the work.

*********
*********

Weaknesses

W1:​​ This paper proposes a novel continual learning paradigm with detailed theoretical derivations. However, I have doubts whether this pipeline approach, which first trains the model based on the new task and then solves the closed form solution to find the optimal **α**, can outperform traditional one-stage incremental training method. Intuitively, more steps lead to error propagation, which may not guarantee improved performance.

W2: Another concern is that the proposed linear interpolation method is overly rigid. Specifically, assigning a same “α” across all network parameters ignores the fact that different parameters contribute unequally to task performance.

​​W3:​​ The derivation in the second part (train with perturbation) seems to be a relatively simple deduction, which lacks explicit linkage to the optimal **α** in the first step. The final conclusion is also somewhat simple, merely introducing some perturbations without clearly establishing a connection to the eventual interpolation method.

​​W4:​​ This paper provides extensive derivations and makes some assumptions. Are these assumptions supported by empirical evidence, such as statistics of parameters from training datasets?

---

> ### Author Rebuttal · Authors · 2025-07-31
>
> We sincerely thank the reviewer for their valuable time and insightful feedback. Below we address the raised concerns point-by-point.
>
> ---
>
> > W1 & Q1: Whether our two-stage pipeline offers clear advantages over traditional one-stage training and  further justification of its necessity and effectiveness.
>
>
> We would like to clarify that our two-stage framework aims at introducing the *model merging* inference paradigm into continual learning (CL) to better preserve prior task knowledge.
>
> Our motivation stems from a key observation: in traditional CL methods, the model trained on the current task is directly used to perform inference on both new and old tasks. This often leads to severe forgetting of previously acquired knowledge. In contrast, our method introduces a **theoretically grounded post-training merging step**. Specifically, we compute a convex combination of the previous inference model $\theta_{t-1}$ and the current task-specific optimum, where the merging coefficient $\alpha_t^*$ is derived in closed form (see Eq. (8)) to explicitly minimize the **total loss increase across all seen tasks**. Importantly, traditional CL approaches are equivalent to setting $\alpha = 1$ in our formulation—thus our method **generalizes and improves upon one-stage training** under the merging paradigm.
>
> Regarding the reviewer’s concern on potential *error propagation* due to a multi-step process, we emphasize that our framework does **not** introduce additional backward passes. Structurally, our "Train + Merge Inference" pipeline is equivalent to standard "Train + Inference" approaches. The only difference is that we apply a **theory-informed model merging step before inference** (i.e., LoRA-M), which empirically **reduces interference from new tasks** and thereby *mitigates* error propagation. As shown in **Figure 1** and **Table 4**, this leads to significant reduction in forgetting while preserving new task plasticity.
>
> ---
>
> > W2: The use of a uniform merging coefficient α across all parameters may be too restrictive.
>
> Using a more fine-grained merging strategy for model parameters may potentially lead to a higher performance upper bound. However, such strategies would introduce additional computational overhead and theoretical complexity. Our experimental results also demonstrate that a unified merging coefficient already achieves strong and competitive performance.
>
> We ran experiments comparing the global α with a more granular strategy—learning separate α values for each LoRA module. The results are as follows:
>
> | ImageNet-R         | 5 tasks | 10 tasks | 20 tasks |
> |--------------------|---------|----------|----------|
> | Per-module α       | 80.53   | 76.82    | 74.68    |
> | Global α (ours)    | 80.88   | 78.48    | 74.13    |
>
> The performance differences are negligible. Moreover, we observed that the learned per-module α values were very similar to each other, suggesting that a global α is sufficient to capture the merging dynamics. Therefore, we adopt a single-α approach for its simplicity, efficiency, and theoretical tractability.
>
> Based on prior research[1][2], a global α preserves the structural integrity of the task vector $\Delta \theta_t$ , which represents a coherent direction in parameter space. A parameter-wise α may introduce inconsistent scaling, distorting the internal correlations of $\Delta \theta^*_t$ and complicating the derivation of a closed-form solution.
>
>
> ---
>
> > W3 & Q2: The theoretical connection between the two stages of our framework.
>
> There is a direct theoretical link: after deriving the optimal $\alpha_t$, we observed that even optimal merging leads to a total performance drop across tasks, quantified as $\sum_{i=1}^{t} \delta_i(\alpha_t)$ in Eq. 11. The **Train with Perturbation** step is specifically designed to reduce this upper bound during training.
>
> The perturbation is not simply random noise; rather, it is a directional regularization based on $\Delta \theta_t$, approximating the Hessian-based quadratic penalty. This encourages convergence to flatter minima that are more robust to parameter interpolation, thus improving the final merged model’s generalization. This effect is clearly visualized in Figure 2, where the merged models from P&M exhibit significantly flatter and wider low-loss regions compared to the baseline.
>
> ---
>
> > W4 & Q3: Whether our theoretical assumptions are empirically grounded and suggests including parameter statistics to support their validity.
>
>
> We would like to clarify that the assumptions we adopt—such as approximating the Hessian with the empirical Fisher Information Matrix, applying a second-order Taylor expansion of the loss, and neglecting higher-order terms—are not unique to our work. These are widely used and well-accepted approximations in the continual learning [3][4][5].
>
> Such approximations have been extensively validated and empirically supported in recent deep learning theory and algorithms. Our use of them follows established practices rather than introducing new or unverified assumptions.
>
>
> ---
>
> We hope that our clarifications and empirical justifications effectively address the reviewer’s concerns.
>
>
> ---
>
> [1] Gabriel Ilharco, et al. "Editing Models with Task Arithmetic." ICLR23.
>
> [2] Hongkang Li, et al. "When is Task Vector Provably Effective for Model Editing? A Generalization Analysis of Nonlinear Transformers." ICLR25.
>
> [3] James Kirkpatrick, et al. "Overcoming Catastrophic Forgetting in Neural Networks." 2017.
>
> [4] Zhenyi Wang, et al. "A UNIFIED AND GENERAL FRAMEWORK FOR CONTINUAL LEARNING." ICLR24.
>
> [5] Mei Li, et al. "BECAME: BayEsian Continual Learning with Adaptive Model MErging." ICML25.

---

> > ### Comment · Reviewer_Fr4E · 2025-08-05
> >
> > Your reply addressed my concerns.
> > I'm willing to raise my score.

---

> > > ### Author Response · Authors · 2025-08-06
> > >
> > > Dear Reviewer Fr4E,
> > >
> > > Thank you very much for your thoughtful reconsideration and for raising your score. ﻿
> > >
> > > We will make sure to incorporate the key clarifications and experiment from this discussion into the revised manuscript.
> > >
> > > Best regards,
> > >
> > > Authors

---

### Note · Authors · 2025-08-12

Dear NeurIPS 2025 AC and reviewers,
﻿
We thank the AC and reviewers for their time and effort. We would like to briefly summarize our discussions and the strengths of our work.

For **Reviewer NW1X**, we provided theoretical and empirical evidence showing that post-hoc merging reduces forgetting and improves FAA, experiments confirming that a global α performs sufficiently well, and a detailed explanation of how perturbation training optimizes the residual bound after optimal α. Reviewer NW1X confirmed **their concerns have been addressed**.

For **Reviewer AEtr**, we showed that memory overhead is modest with diagonal Fisher for LoRA modules, added experiments against strong baselines, and demonstrated that perturbation training benefits merging-based methods. Reviewer AEtr confirmed that **all concerns have been addressed**.

For **Reviewer 47Vz**, we explained theoretical and practical differences from EWC, confirmed low memory usage with diagonal Fisher, and tested an online P&M variant without storing all Fishers. Reviewer 47Vz acknowledged that **our clarifications and experiments addressed their concerns**.

For **Reviewer VKzv**, we justified our assumptions with standard continual learning practices and ablation studies, expanded related work and baselines, clarified the method’s applicability beyond LoRA, and committed to improving visualization explanations. Reviewer VKzv **appreciated our clarifications and additional experiments**.

Overall, we propose **P&M**, a principled continual learning framework that unifies perturbation-based training with post-hoc parameter merging to mitigate catastrophic forgetting. By deriving a closed-form optimal merging coefficient from a total-loss minimization objective and introducing task-vector-aligned stochastic perturbations during training, P&M reduces parameter interference while enhancing generalization. Integrated with LoRA for memory efficiency, P&M achieves consistent and significant gains across diverse CL benchmarks.
﻿

We sincerely thank the AC and reviewers once again for their time, constructive feedback, and valuable discussions throughout the review process.

Best regards,

Authors

---

### Decision · Program_Chairs · 2025-09-17

**Decision:**

Accept (poster)

**Comment:**

The paper received all positive reviews, leading to a final acceptance recommendation.